# On the Transfer of Object-Centric Representation Learning

**Aniket Didolkar**[1,*]   **Andrii Zadaianchuk**[2]   **Anirudh Goyal**[1]   **Mike C. Mozer**[3]
**Yoshua Bengio**[1]   **Georg Martius**[4]   **Maximilian Seitzer**[4,*]

[1]MILA & University of Montreal      [2]University of Amsterdam
[3]Google Deepmind      [4]MPI for Intelligent Systems & University of Tübingen

## Abstract

The goal of object-centric representation learning is to decompose visual scenes into a structured representation that isolates the entities into individual vectors. Recent successes have shown that object-centric representation learning can be scaled to real-world scenes by utilizing features from pre-trained foundation models like DINO. However, so far, these object-centric methods have mostly been applied in-distribution, with models trained and evaluated on the same dataset. This is in contrast to the underlying foundation models, which have been shown to be applicable to a wide range of data and tasks. Thus, in this work, we answer the question of whether current real-world capable object-centric methods exhibit similar levels of transferability by introducing a benchmark comprising seven different synthetic and real-world datasets. We analyze the factors influencing performance under transfer and find that training on complex natural images improves generalization to unseen scenarios. Furthermore, inspired by the success of task-specific fine-tuning in foundation models, we introduce a novel fine-tuning strategy to adapt pre-trained vision encoders for the task of object discovery. We find that the proposed approach results in state-of-the-art performance for unsupervised object discovery, exhibiting strong zero-shot transfer to unseen datasets.

## 1 Introduction

In the past decade, deep learning-based approaches have become ever more general, culminating in models that exhibit broad and flexible vision (Dehghani et al., 2023; Oquab et al., 2023) and language understanding (Brown et al., 2020; OpenAI Team, 2024). These so-called foundation models can be applied to a variety of data and tasks in a zero-shot manner. An open challenge is how to equip these models with the ability to robustly reason about visual inputs, that is, in a manner that supports compositional generalization and causal reasoning (Schölkopf et al., 2021; Goyal & Bengio, 2022). Evidence suggests that human cognition deals with these problems by dynamically binding raw perceptual features into symbol-like entities that can be flexibly composed together and reasoned over (Pinker, 1984; Spelke, 1990; 2000). Inspired by these findings, the field of *object-centric representation learning* aims to replicate these abilities in deep learning models (Greff et al., 2020). By mirroring the compositional generative process of the world (Brady et al., 2023), these methods learn to decompose visual scenes into structured representations capturing the objects in the scene in a fully unsupervised way. Not only can object-centric representations provably exhibit compositional generalization (Wiedemer et al., 2024), they also support a diverse set of downstream tasks such as world modeling (Ke et al., 2021; Wu et al., 2023a), robotic control (Zadaianchuk et al., 2020; Driess et al., 2023; Didolkar et al., 2024; Haramati et al., 2024), visual question answering (Xu et al., 2024; Mamaghan et al., 2024), and compositional generation in 2D (Singh et al., 2022a; Wu et al., 2023b; Jiang et al., 2023) and 3D (Sajjadi et al., 2022; Jabri et al., 2023).

---

*equal contribution. Correspondence: `maximilian.seitzer@tue.mpg.de`, `adidolkar123@gmail.com`.
Website: `rw-ocrl.github.io/ftdinosaur-paper`

While long confined to simplistic synthetic datasets (Eslami et al., 2016; Greff et al., 2019; Engelcke et al., 2020; Locatello et al., 2020), recent progress has scaled object-centric representations to complex real-world image (Seitzer et al., 2023; Wu et al., 2023b; Jiang et al., 2023; Kakogeorgiou et al., 2024; Löwe et al., 2024) and video datasets (Zadaianchuk et al., 2023b; Aydemir et al., 2023). This success is enabled by the use of pre-trained features from self-supervised vision encoders such as DINO (Caron et al., 2021). While it has been shown that the representations produced by such self-supervised models are fairly robust to changes to the training distribution, it is unclear *whether the same holds for an object-centric model trained on top of such representations*. Thus, in this work, we study the question of how well object-centric representations using pre-trained features transfer to new data. This comprises (1) the ability of models to discover objects in unseen scenarios and (2) the robustness of the object representation itself. In particular, we focus on the "zero-shot" setting, where models are presented with object categories never seen during training. This setting is relevant as a model with strong zero-shot abilities could serve as a "foundation model" for object-centric representations.

To this end, we introduce a benchmark consisting of 7 datasets comprising a diverse range of synthetic and real-world scenes. Using this benchmark, we (1) seek to understand the zero-shot transfer capabilities of existing models, and (2) study the properties of training datasets that influence generalization. The general conclusion we draw from this benchmark is that object-centric models which are trained on naturalistic datasets consisting a variety of objects — such as COCO (Lin et al., 2014) — usually exhibit decent zero-shot generalization.

Equipped with this knowledge, we aim to build a strong general-purpose object-centric model. To achieve this, we first make the observation that current approaches for real-world object-centric learning (Seitzer et al., 2023; Wu et al., 2023b; Jiang et al., 2023; Zadaianchuk et al., 2023b; Aydemir et al., 2023) use *fixed* pre-trained encoders (e.g. with the DINO method (Caron et al., 2021)) to encode the input. This may be limiting as, while the pre-trained encoders offer good general-purpose features, they may not be optimal for the task of object discovery. Instead, we propose to finetune the encoder parameters for the target task; to this end, we introduce a suitable training recipe as well as a novel decoder that reduces the increased computational costs from finetuning. Building on the DINOSAUR model (Seitzer et al., 2023), our proposed finetuning approach sets a new state-of-the-art for real-world object-centric learning on the COCO dataset, as well as in the zero-shot setting. Our method shows zero-shot transfer across a multitude of diverse datasets, often achieving and even surpassing the in-distribution performance on these datasets.

Our contributions are as follows:

- We introduce a benchmark to evaluate the transfer of real-world capable object-centric learning methods (Sec. 3.1).
- Using the benchmark, we analyze the zero-shot capabilities of object-centric models (Sec. 3.2) and investigate dataset properties for training generalizable models (Sec. 3.3).
- We propose a finetuning approach applied to DINOSAUR, which allows the stable adaptation of the parameters of the pre-trained encoder for the task of object discovery (Sec. 4).
- The resulting method, FT-DINOSAUR, achieves state-of-the-art results across various in-distribution and out-of-distribution scenarios (Sec. 5).

## 2 RELATED WORK

**Object-Centric Learning on Real-World Datasets**    Originally, object-centric methods were mostly applied to synthetic data with limited complexity (Johnson et al., 2017; Karazija et al., 2021; Greff et al., 2022) and trained from scratch (Eslami et al., 2016; Burgess et al., 2019; Lin et al., 2020; Locatello et al., 2020; Traub et al., 2023). Recently, there has been considerable interest (Elsayed et al., 2022; Singh et al., 2022b; Seitzer et al., 2023; Zadaianchuk et al., 2023b; Wu et al., 2023b; Didolkar et al., 2024; Löwe et al., 2024; Kakogeorgiou et al., 2024) in scaling those methods to complex and unconstrained real-world image and video datasets like COCO (Lin et al., 2014) or YouTube-VIS (Yang et al., 2019). Current state-of-the-art techniques (Seitzer et al., 2023; Wu et al., 2023b; Jiang et al., 2023; Kakogeorgiou et al., 2024; Aydemir et al., 2023; Zadaianchuk et al., 2023b) rely on applying slot attention (Locatello et al., 2020) to frozen vision transformers (ViT) (Dosovitskiy et al., 2021) pre-trained with contemporary self-supervised representation learning methods (Caron et al., 2021; Oquab et al., 2023). Approaches differ by their learning objective; one line of models

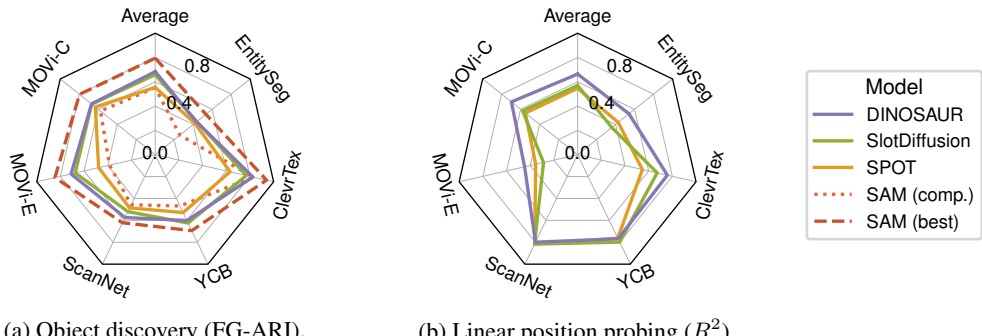

(a) Object discovery (FG-ARI).   (b) Linear position probing ($R^2$).

**Figure 1**: **Evaluating zero-shot transfer of current object-centric models** trained on the COCO dataset. **(a)**: object discovery performance in terms of FG-ARI. **(b)**: linear probing of object position from slots in terms of $R^2$ score. We report results for mBO and category probing in Fig. A.1.

is based on DINOSAUR (Seitzer et al., 2023) and utilizes a feature reconstruction objective (Seitzer et al., 2023; Zadaianchuk et al., 2023b; Kakogeorgiou et al., 2024; Aydemir et al., 2023), whereas others apply diffusion objectives (Jiang et al., 2023; Wu et al., 2023b). Although these techniques confirm that object-centric representation learning *is* possible for complex real-world inputs, they are also limited by the quality of self-supervised encoders, as those encoders remain frozen during object-centric training. In contrast, our method, while starting from self-supervised features, adapts them through object-centric finetuning, making them more suitable for the task of object-centric scene decomposition.

**Zero-shot Transfer**  A paradigm first introduced by Larochelle et al. (2008), zero-shot transfer is the application of models to tasks or datasets not seen during training. Recently, zero-shot transfer has become more prevalent in deep learning due to the availability of large foundation models (Oquab et al., 2023; Kirillov et al., 2023; OpenAI Team, 2024); these models utilize large-scale pre-training to develop robust general-purpose abilities that can be applied zero-shot to various tasks and datasets. In the context of object-centric learning, model transfer has been studied by testing models under various types of distribution shifts, changing factors like object color, texture, or the number of objects (Karazija et al., 2021; Dittadi et al., 2022; Wiedemer et al., 2024). Generally, these works only change a single factor of variation between training and test data. However, in real-world settings, distribution changes are much more uncontrolled than this: for instance, transferring to a novel category of object might include (1) multiple simultaneous factor changes, and (2) *completely new* factors of variation. Thus, in this work, we study this more challenging setting by evaluating the zero-shot transfer to arbitrary object categories.

**Task-Specific Finetuning**  While pre-training models on large and diverse datasets often offers good baseline performance for zero-shot transfer (Radford et al., 2019; Kirillov et al., 2023; Oquab et al., 2023), further improvement can be achieved by finetuning the model for the target task. For example, self-supervised vision representations (Caron et al., 2021) can be adapted by finetuning for the tasks of unsupervised semantic segmentation (Hamilton et al., 2022; Zadaianchuk et al., 2023a) or multi-object tracking in videos (Salehi et al., 2023; Tumanyan et al., 2024). In this work, we similarly propose to adapt pre-trained vision representations by finetuning them for the task of object discovery.

## 3 WHAT MATTERS FOR TRANSFER OF OBJECT-CENTRIC REPRESENTATIONS?

In this section, our goal is to (1) measure the degree to which real-world capable object-centric models support zero-shot transfer, and (2) understand the factors influencing the models' transfer ability. For (1), we introduce a benchmark for measuring zero-shot performance in Sec. 3.1 and compare models in Sec. 3.2. For (2), we study the impact of training data on transfer in Sec. 3.3.

### 3.1 BENCHMARK FOR ZERO-SHOT TRANSFER

Our goal is to test the zero-shot transfer of real-world capable object-centric models utilizing pre-trained features. As this class of model is typically trained on the COCO dataset (Lin et al., 2014), we also use COCO as the training dataset for our benchmark. For our benchmark, we interpret zero-shot transfer to mean that a model can successfully discover objects of novel *categories* not occurring in the training data. Objects of different categories (e.g. humans; cars) have at least one factor of variation that is not shared between them (e.g. eye color; number of wheels); thus, we use "category" as a criterion to make sure there is sufficient difference between train and test sets.[1]

**Evaluation Datasets**   To obtain a test bed that robustly measures zero-shot performance, we gather the evaluation splits of several datasets previously proposed by the object-centric community, with diverse properties and increasing complexity: CLEVRTEX (Karazija et al., 2021), SCANNET and YCB as used in Yang & Yang (2022), and MOVi-C and MOVi-E (Greff et al., 2022). Additionally, we add the challenging ENTITYSEG dataset (Lu et al., 2023), consisting of in-the-wild real-world images with high-quality mask annotations. While ENTITYSEG includes images with categories also occurring in the COCO dataset, it is an open world dataset without pre-defined classes; thus, we consider it adequate for the purpose of this benchmark. In total, we gathered 6 datasets with a total of 18 874 images. For analysis, we also use the PASCAL VOC (Everingham et al., 2012) dataset, but do not include it in the zero-shot benchmark as its set of categories is fully included in the COCO dataset used for training. For further details on the datasets, we refer to App. E.

**Metrics**   Similar to prior work (e.g. Dittadi et al., 2022), we evaluate the object representation in terms of object discovery (do the masks associated with each object align with the true object masks?) and downstream property prediction (are the representations informative about the objects?). To aggregate results over different datasets, we compute the *per-sample average*, normalizing by the dataset size (see Table E.9). We now provide a brief overview of the metrics; see App. F for details.

For *object discovery*, we compute the commonly used *foreground ARI* (FG-ARI) (Rand, 1971; Hubert & Arabie, 1985), measuring how well the discovered objects follow the separation prescribed by the reference mask annotations. In addition, we compute the *mean best overlap* (mBO) (Pont-Tuset et al., 2017), measuring how well the discovered masks fit to objects. To measure (in-distribution) performance on the COCO dataset, we additionally evaluate *panoptic scene decomposition* into "things" (objects) and "stuff" (background). This is sensible because on real-world images, there is no clear distinction between objects and background from the model's point-of-view. To this end, we compute *panoptic ARI* (P-ARI) and *class-agnostic panoptic quality* (PQ), where the latter measures both mask quality and precision/recall (Kirillov et al., 2019). For *downstream property prediction*, we closely stick to the protocol of Dittadi et al. (2022). In particular, we train linear predictors (probes) on top of the object representation to predict position and category (where available) of the objects. For position, we use the $R^2$ score as a metric; for classification, we use accuracy.

### 3.2 EVALUATING MODELS

**Models**   We evaluate three recent methods capable of real-world object-centric learning, all using pre-trained DINO features as inputs:DINOSAUR (Seitzer et al., 2023), which also uses DINO features as targets; SPOT (Kakogeorgiou et al., 2024), which builds upon DINOSAUR by improving the decoder; and SlotDiffusion (Wu et al., 2023b), which uses a diffusion decoder. We limit ourselves to pre-trained feature-based methods as other approaches have not shown scalability to real-world data. To showcase the gap between state-of-the-art supervised and unsupervised methods for object discovery, we evaluate the Segment Anything model (SAM) (Kirillov et al., 2023), a supervised segmentation foundation model known for its zero-shot capabilities. We use two configurations: for SAM (best), we use the largest available model (ViT-Huge), and pick the mask confidence threshold resulting in the best performance per-dataset; in contrast to current object-centric methods, this results in a variable number of masks per-image. For better comparability, we also evaluate a baseline SAM (comp.), using a ViT-Base encoder and a fixed number of masks. Please refer to App. D for details about the models.

---

[1]Note that for our purposes, we are not going to take into account the data the utilized self-supervised pre-trained models (e.g. DINO) were trained on (e.g. ImageNet). This is because our primary interest is in the transfer of the object-centric components, which are trained on a distinct set of images (e.g. COCO).

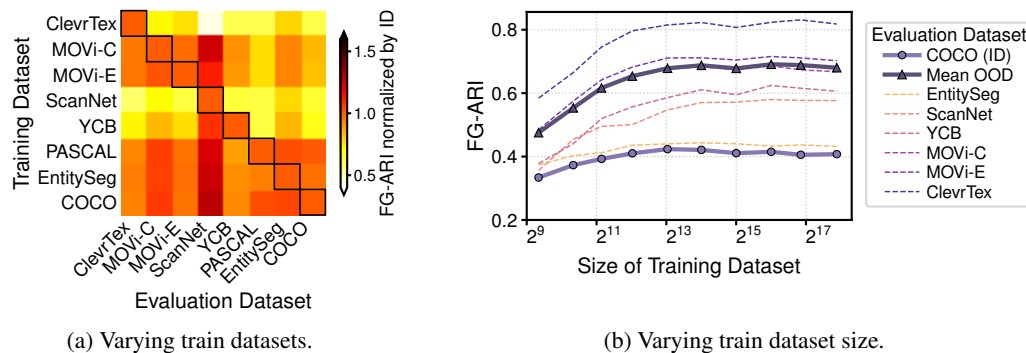

(a) Varying train datasets.

(b) Varying train dataset size.

**Figure 2**: **Effect of training data on transfer of the DINOSAUR model.** Performance in FG-ARI, see Fig. A.2 for corresponding plots with mBO. **(a)**: comparing transfer to in-distribution performance for different training datasets. **(b)**: scaling behavior for training on differently sized subsets of COCO.

**Results**    We present the results in Fig. 1 and Fig. A.1, with example predictions in App. G. For *object discovery*, DINOSAUR and SlotDiffusion are on par in terms of FG-ARI, while SlotDiffusion achieves the best mBO. Unsurprisingly, SAM (best) outperforms the unsupervised methods; this is especially pronounced in terms of mBO — the difference can be explained by SAM's ability to output a variable number of masks, which allows SAM to discover smaller objects not captured by methods with a fixed number of slots. If we remove that privilege, the SAM (comp.) baseline is inferior to the unsupervised models on many datasets, with the exception of CLEVRTEX. For *downstream property prediction*, we find DINOSAUR to perform best on average, allowing for a decent property readout on most datasets. We also evaluate property prediction for DINOSAUR models trained in-distribution on each dataset. Comparing the average performance of the zero-shot model to the in-distribution models, we see similar results for predicting categories (+1% rel. change), and only slight drops for predicting positions (-11% rel. change). We conclude that object-centric models already exhibit decent zero-shot generalization to unseen datasets.

## 3.3    EVALUATING TRAINING DATA

So far, we used the same training data for comparing the models — a natural question is how the *training data* affects transfer behavior. To answer it, we train DINOSAUR on different training datasets and evaluate the zero-shot performance in terms of object discovery. We organize our experiments into two groups: (1) varying the data distribution, and (2) varying the amount of samples from a particular data distribution. The former allows us to identify properties of the data that influence zero-shot behavior, whereas the latter investigates how models scale with data.

**Properties of the Data Distribution**    To obtain training datasets with different properties, we utilize the training splits belonging to the benchmark datasets listed in Sec. 3.1. We characterize the training datasets along three dimensions: **realism** — in terms of the three categories synthetic (CLEVRTEX), hybrid (MOVi, SCANNET, YCB) and natural (PASCAL VOC, COCO, ENTITYSEG); **diversity** — on a spectrum from narrow to broad (roughly CLEVRTEX ≪ SCANNET, YCB, PASCAL VOC ≪ MOVi ≪ COCO ≪ ENTITYSEG); and the **amount of objects** — ranging from few (PASCAL VOC) to moderate (up to 6; CLEVRTEX, SCANNET, YCB) to many (MOVi, COCO, ENTITYSEG). The results are shown in Fig. 2a.

First, we find that training and evaluating in-distribution, i.e. on matching datasets, unsurprisingly performs best in general. Training on *synthetic* and *hybrid* datasets transfers well to datasets in those categories, but not to natural data; conversely, *natural* data transfers well to synthetic and hybrid data. Next, we find that the zero-shot performance is fairly similar when trained on COCO and ENTITYSEG (high diversity, many objects), or PASCAL VOC (less diversity, few objects). This shows that having complex natural data is more important for zero-shot performance compared to data diversity. Moreover, even when trained on natural data with few objects (e.g. PASCAL VOC), the model transfers well to datasets with more objects such as MOVi, COCO, and ENTITYSEG.

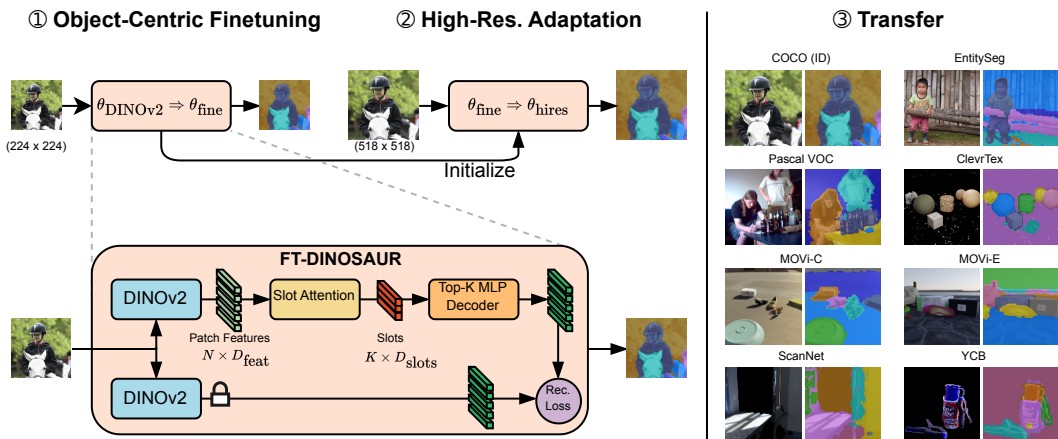

**Figure 3**: **Overview of our method "FT-DINOSAUR".** ① **Object-Centric Finetuning**: starting from DINOv2, the encoder is finetuned for the task of object discovery on the COCO dataset. ② **High-Res Adaptation**: the model is further adapted to high-resolution images. ③ **Transfer**: at test time, we transfer the trained model to different datasets from our proposed benchmark (Sec. 3.1).

Overall, we can conclude that training on natural data leads to strong zero-shot performance of current object-centric models.

**Effect of Data Scale**   We now investigate the effect of the number of training data points. To do so, we train DINOSAUR on differently sized subsets of the COCO dataset (up to 240k samples when including the *"unlabeled"* split). From the results in Fig. 2b, we find that in-distribution performance plateaus around $8\,192$ ($2^{13}$) samples, and zero-shot performance around $16\,182$ ($2^{14}$) samples. Intriguingly, this shows that current object-centric models can be very sample efficient in obtaining decent in-distribution and competitive zero-shot generalization. However, we do not find evidence of favorable data scaling laws.

### 3.4   SUMMARY

Our initial question was whether object-centric models based on pre-trained models inherit some of their generality. Our results show that this is indeed the case, as the models we tested (DINOSAUR, SPOT, SlotDiffusion) exhibit decent zero-shot transfer to unseen object categories. Furthermore, our experiments show that training on complex natural data is an important component for zero-shot transfer, which can be attributed to the inherent complexities associated with such data. In addition, real-world datasets offer a significantly larger catalog of objects and instances to train on compared to synthetic or hybrid datasets.

Equipped with this knowledge, we shift our focus to enhancing the performance of unsupervised object-centric models. Specifically, the question we ask is: *Can we improve object-centric representations by finetuning pre-trained encoders for the task of object discovery?*

## 4   OBJECT-CENTRIC FINETUNING

Current methods for real-world object-centric learning (Seitzer et al., 2023; Jiang et al., 2023; Wu et al., 2023b; Kakogeorgiou et al., 2024; Zadaianchuk et al., 2023b; Aydemir et al., 2023; Löwe et al., 2024) are all based on pre-trained self-supervised features (Caron et al., 2021; Oquab et al., 2023). While those features offer good performance for many downstream tasks out of the box, they are not explicitly designed for the *task of object discovery*. We conjecture that this gap between training and downstream objective leads to sub-optimal transfer performance. Thus, we propose to adapt the pre-trained features by *task-specific finetuning* — Fig. 3 shows our approach.

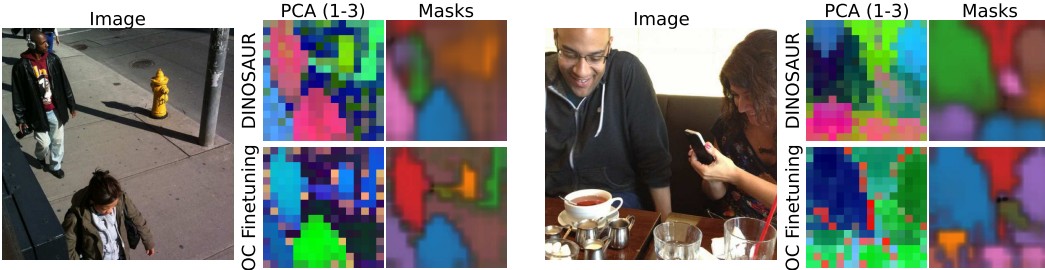

**Figure 4**: **Visualization of encoder features in DINOSAUR (frozen DINOv2 features) and for features adapted with object-centric finetuning.** We show the 1st to 3rd PCA components visualized by different RGB channels (second column). The last column shows scene decomposition masks by each method. More examples and additional PCA components are shown in Fig. A.4.

## 4.1 FINETUNED DINOSAUR

**Finetuning** We first describe how we adapt the DINOSAUR architecture (Seitzer et al., 2023) for finetuning. DINOSAUR uses a pre-trained ViT as the encoder that is kept fixed during training. The original work reported that unfreezing the encoder leads to a collapse; this is because the encoder features are simultaneously used as the model's prediction targets. To sidestep this problem, we add a *target encoder* that is initialized to be a copy of the original encoder, but kept fixed throughout training. This allows us train the full model end-to-end without collapse.

We found that the encoder would initially drift away from its pre-trained initialization, likely induced by the noisy gradients from the randomly initialized slot attention module. To reduce the effect of this, we introduce blockwise exponentially decaying learning rates (Howard & Ruder, 2018) for the encoder. Furthermore, we found a better set of hyperparameters, namely a lower learning rate, switching to a cosine learning rate schedule (Loshchilov & Hutter, 2017), lower gradient clipping, weight decay on the encoder and a higher batch size. We detail the exact settings in App. C.4. The efficacy of this improved setup is shown by the fact that we can now also train the model starting from a *randomly initialized* ViT encoder (42.3 FG-ARI, 27.3 mBO), a scenario which was previously reported as leading to collapse by Seitzer et al. (2023). We further expand on this setup in Section A. We also experimented with an EMA student-teacher setup to continuously adapt the targets throughout training, but found that this leads to worse results (see App. A.2).

**High Resolution Adaptation** A further way to make more effective usage of data is to increase the image resolution. Standard ViTs use a relatively low resolution of $224 \times 224$ pixels, leading to a patch resolution of $16 \times 16$ when trained with patch size 14. This hides details, inhibits capturing smaller objects, and leads to coarser objects masks. Thus, after training at $224 \times 224$ resolution, we add a short second stage of training, in which the model is adapted to image resolution to $518 \times 518$ (i.e. $37 \times 37$ patches) over $10\,000$ steps. This is similar DINOv2's training strategy (Oquab et al., 2023), and adds significant improvements (Sec. 4.3) without a high computational burden.

**Efficient Top-K Decoding** Finetuning the encoder and high resolution adaptation both significantly increase the costs in terms of computation and memory. To mitigate this, we introduce a novel efficient decoding approach based on the MLP decoder introduced by Seitzer et al. (2023), which we call *top-k decoding*. For each of $N$ patches, the MLP decoder produces an output by combining the predictions over $C$ slots using a slot-wise weighted average, resulting in a computational cost of $\mathcal{O}(N \cdot C)$. Our insight is that *most of this computation is wasted*, as slots are localized and mostly sparsely distributed across the image — instead, it suffices to decode the $k$ *most likely slots* occupying a patch, reducing the costs to $\mathcal{O}(N)$ for constant $k$. While we do not have access to the true occupation probabilities apriori, empirically we found that the masks from slot attention can serve as a good proxy. We refer to App. C.3 for more details.

**Table 1**: **Ablation study on COCO.** Starting from DINOSAUR (Seitzer et al., 2023) (first row), we ablate the impact of switching to DINOv2, finetuning the encoder (FT), improving general (G-HP) and encoder hyperparameters (E-HP), adding top-k decoding and high-resolution adaptation. Results averaged over 3 random seeds besides last two rows, which use 5 seeds.

| Model | FT | G-HP | E-HP | FG-ARI | mBO | P-ARI | PQ |
|---|---|---|---|---|---|---|---|
| DINO ViT-B/16 | ✗ | ✗ | ✗ | 40.3 | 27.2 | 37.1 | 14.4 |
| DINOv2 ViT-S/14 | ✗ | ✗ | ✗ | 42.5 | 28.8 | 39.5 | 16.3 |
| | ✗ | ✓ | ✗ | 42.9 | 29.1 | 39.8 | 16.8 |
| | ✓ | ✗ | ✗ | 46.5 | 29.8 | 42.2 | 17.9 |
| | ✓ | ✓ | ✗ | 48.0 | 30.6 | 42.8 | 18.8 |
| | ✓ | ✓ | ✓ | 48.5 | 30.7 | 42.6 | 19.0 |
| +Top-k | ✓ | ✓ | ✓ | 46.4 | 32.0 | 43.5 | 19.5 |
| +Top-k, +Hi-Res | ✓ | ✓ | ✓ | 46.6 | 35.6 | 49.6 | 23.6 |

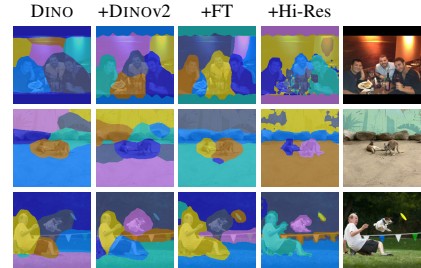

## 4.2 ANALYSIS

Object-centric finetuning adapts the pre-trained encoder such that the original DINOv2 features can be predicted better, with the slot representations acting as a bottleneck. To better understand the effect of this procedure, we study how the encoder representations change after finetuning. In Fig. 4 and Fig. A.4, we show the first PCA components obtained from DINOv2 features (used by DINOSAUR) and from features after object-centric finetuning. DINOv2 features mainly exhibit semantic similarity, i.e. one component often corresponds to several different objects or parts of the same category (such as human heads). In contrast, after object-centric finetuning, PCA components are noticeably object-centric, splitting instances of the same category and grouping together different object parts into one component. To confirm this observation quantitatively, we apply per-image k-means clustering to the two types of features. On COCO, we find that the clustering of features from object-centric finetuning corresponds better to object instances, reaching 34.0 FG-ARI and 28.7 mBO in contrast to 27.4 FG-ARI and 24.7 mBO for the original DINOv2 features.

## 4.3 ABLATIONS

In Table 1, we analyze the contribution of different components of our model on the COCO dataset, starting from the original DINOSAUR model and ending with our final model. First, we find that *switching from DINO to DINOv2* leads to moderate improvements (+2.2 FG-ARI and +1.6 mBO). Adding *finetuning* results in a strong improvement of FG-ARI (+4.0), demonstrating the importance of task-specific adaptation. To evaluate our *hyperparameter changes*, we split them into two groups: general hyperparameters (cosine schedule, lower learning rate, lower gradient clipping), and encoder hyperparameters (blockwise learning rates, lower encoder learning rate, encoder weight decay). The changes to the general hyperparameters result in moderate improvements (+1.5 FG-ARI, +0.8 mBO, +0.9 PQ), with the changed encoder hyperparameters contributing further small improvements. Introducing *top-k decoding* reduces FG-ARI (-2.1), but increases the other metrics (e.g. +1.3 mBO). Finally, *high-resolution adaptation* results in further strong boosts (+3.6 mBO, +6.1 P-ARI, +4.1 PQ).

## 5 EVALUATION

To evaluate our approach, we use our benchmark to answer the following three questions:

- How does our proposed finetuning methodology work on diverse datasets (Sec. 5.1)?
- How does our method compare to prior methods for real-world object-centric learning (Sec. 5.2)?
- How does our method perform on the introduced zero-shot benchmark (Sec. 5.3)?

## 5.1 EVALUATION OF OBJECT-CENTRIC FINETUNING

We first validate our proposed finetuning approach as a general methodology by training on diverse datasets. In particular, we train a DINOSAUR model using a DINOv2 backbone with and without

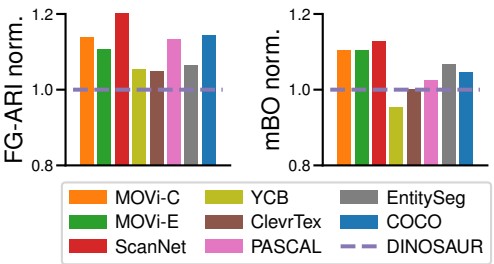

**Figure 5**: **Normalized performance when adding *finetuning* to DINOSAUR for *in-distribution* training**, using a ViT-S/14 DINOv2 encoder. Finetuning shows strong gains on all datasets. Numerical results in Table A.4.

| Model | FG-ARI | mBO | P-ARI | PQ |
|---|---|---|---|---|
| DINOSAUR | 40.5 | 27.7 | 37.1 | 14.4 |
| SlotDiffusion† | 37.3 | 31.4 | 47.6 | 21.0 |
| SPOT † | 37.0 | 34.8 | **52.4** | 21.3 |
| FT-DINOSAUR | **48.8** | **36.3** | 49.4 | **23.9** |
| SAM *(comp.)*† | 12.1 | 19.0 | 10.8 | 9.4 |
| SAM *(best.)*† | 44.9 | 56.9 | 54.4 | 10.9 |

**Table 2**: **Comparison to prior work on COCO.** We use a ViT-B/14 encoder with top-k decoding and hi-res. adaptation. Results for (FT-)DINOSAUR averaged over 3 seeds. Results marked † evaluate official checkpoints, supervised models in gray. We compare to more baselines in Table A.5.

finetuning on the training splits of the datasets we used for analysis in Sec. 3.3, and evaluate *in-distribution*. The results are listed in Fig. 5. We find that adding finetuning results in strong improvements on both FG-ARI (up to 11 points) and mBO (up to 5 points) across all 8 datasets. First, this demonstrates that finetuning, when using our training recipe, is a general strategy to improve performance of slot attention-based object-centric models with pre-trained backbones. This is in contrast to Seitzer et al. (2023)'s findings, who reported collapsing slots when finetuning the pre-trained ViT encoder. Second, this shows that while pre-trained features obtained from self-supervised methods like DINOv2 are powerful, it is possible to improve upon them with task-specific finetuning. Interestingly, even though the model's objective is to *predict* DINOv2 features, the optimal input to slot attention are *not* those exact features. Following our analysis in Sec. 4.2, we conjecture that finetuning adapts the features to simplify grouping under the inductive biases of the model.

## 5.2 COMPARISON TO PRIOR WORK ON REAL-WORLD OBJECT-CENTRIC LEARNING

Second, in Table 2, we compare our full approach with prior work on the COCO dataset. We find that our method sets a new state-of-the-art on COCO, achieving better results than all previous unsupervised object-centric methods, except being slightly worse than SPOT on the panoptic ARI metric. Moreover, our method also outperforms the SAM *(comp.)* baseline (ViT-Base encoder, same number of masks) on all metrics. In particular, our method has strongly improved FG-ARI (+9), indicating much better object discovery capabilities — it even achieves higher FG-ARI than the SAM *(best.)* baseline (ViT-Huge encoder, variable number of masks). However, there is still a large gap to SAM's mBO, which we attribute to 1) SAM's generally higher mask quality, and 2) its ability to capture a variable number of objects, which in particular leads to finding more small objects. We refer to Fig. G.6 for example predictions.

## 5.3 ZERO-SHOT EVALUATION

Finally, we evaluate our method in terms of its zero-shot performance. First, in Fig. 6, we compare the zero-shot performance of our model finetuned *on* COCO (without top-k decoding or high-res. adaptation) to the performance of our model finetuned *in-distribution*. We find that transferring from COCO yields comparable results to training in-distribution on most datasets (SCANNET, ENTITY-SEG), and even surpasses *in-distribution* training on some datasets (MOVi-C, YCB) — surprisingly, object-centric finetuning does not hurt generalization (e.g. by overfitting), indicating that it adapts the model to the *task* rather than the *data*. Overall, this shows that task-specific finetuning on diverse real-world data is a viable path to obtain zero-shot object-centric models.

Second, in Fig. 7, we compare the zero-shot performance of our full model (including top-k decoding and high-resolution adaptation) to prior work. Averaged over all datasets, our approach achieves both the highest FG-ARI and mBO, while previous work generally trades off high FG-ARI with low mBO (DINOSAUR), or high mBO with low FG-ARI (SlotDiffusion, SPOT). On top of finetuning, we

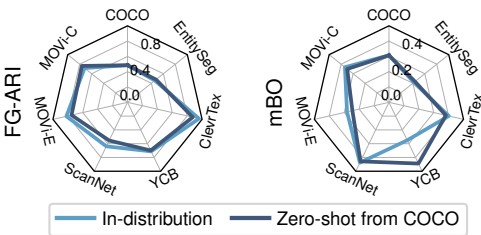
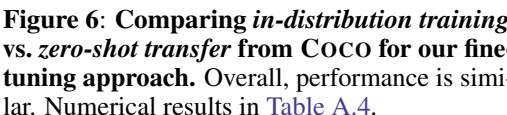
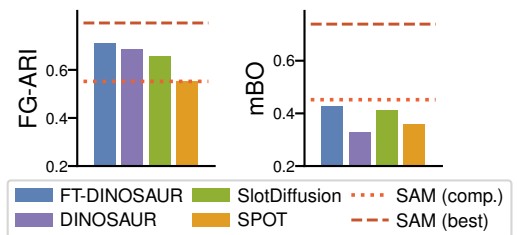

**Figure 6**: **Comparing *in-distribution training* vs. *zero-shot transfer* from COCO for our finetuning approach.** Overall, performance is similar. Numerical results in Table A.4.

**Figure 7**: **Zero-shot performance averaged over datasets.** FT-DINOSAUR performs best both in FG-ARI and mBO. Results per datasets available in Table A.6.

ascribe this to our usage of the MLP decoder (higher FG-ARI) in combination with high-resolution training (higher mBO).

Last, we compare our model to SAM. SAM *(comp.)* generally performs *worse than our model*, showing the difficulty of unsupervised scene decomposition in the absence of task-specific information. For SAM *(best.)*, there is still a large difference to our approach in terms of mBO (30.7 points), while the difference in terms of FG-ARI is much smaller (8.3 points). Taken together, these results show that unsupervised object-centric models are *closing the gap to supervised methods in terms of zero-shot object discovery*. This is astonishing, given that SAM was trained on 10 million images with over 1 billion mask annotations. Furthermore, a principal advantage of object-centric models over SAM is that they come equipped with explicit object representations. While mask quality as measured by mBO is lacking behind SAM, we are hopeful that this gap is addressable by training on even higher resolution images and introducing innovations for variable number of slots. Finally, in the appendix, we present a comparison of the masks obtained from the proposed approach and all baselines (App. G), and comprehensively study failure cases of our model (App. B).

## 6 CONCLUSION

In this work, we have introduced a benchmark of diverse real-world and synthetic datasets to study the zero-shot capabilities of object-centric representation learning models. Our findings indicate that object-centric models using pre-trained encoders already exhibit notable zero-shot capabilities when trained on real-world data. We then presented a finetuning procedure for adapting pre-trained encoders to the task of object discovery, demonstrating that this approach achieves state-of-the-art results across 8 datasets in both in-distribution and out-of-distribution scenarios. We believe that our contributed tools — the zero-shot benchmark and stable finetuning — are important stepping stones towards an object-centric foundation model.

Our benchmark showed the importance of the type of training data for zero-shot transfer. Our experiments indicate that training on complex natural data is important, suggesting an exciting direction to design curated datasets for zero-shot object-centric learning. Moreover, our benchmark revealed that current object-centric models are highly sample-efficient but fail to leverage larger datasets to improve performance at current model sizes. This result is significant because it suggests that, unlike other deep learning domains, stronger object-centric models cannot be achieved simply by scaling up data alone. We hope our findings will encourage the community to develop object-centric models that scale effectively with both data and model size.

Finally, to build general-purpose object-centric models, it is crucial that the learned object-centric representation are useful for real-world downstream tasks. In this work, we evaluated downstream property prediction, which can already serve as a proxy for more complex tasks. While the downstream applicability of object-centric representations has been explored with various tasks — such as reinforcement learning (Yoon et al., 2023), dynamics prediction (Wu et al., 2023a), compositional image generation (Wu et al., 2023b), or visual question answering (Xu et al., 2024; Mamaghan et al., 2024) — the zero-shot scenario has not been comprehensively studied so far. Thus, an exciting direction for future work is to extend our benchmark to include a suite of zero-shot downstream tasks.

# 7 ACKNOWLEDGEMENT

This work was supported by the ERC - 101045454 REAL-RL and funded by EXC number 2064/1 – Project number 390727645. We acknowledge the support from the German Federal Ministry of Education and Research (BMBF) through the Tübingen AI Center (FKZ: 01IS18039B). Andrii Zadaianchuk is funded by the European Union (ERC, EVA, 950086). Views and opinions expressed are, however, those of the author only and do not necessarily reflect those of the European Union or the European Research Council Executive Agency. Neither the European Union nor the granting authority can be held responsible for them. The authors thank the International Max Planck Research School for Intelligent Systems (IMPRS-IS) for supporting Maximilian Seitzer. Aniket Didolkar was partly funded by the UNIQUE Excellence Scholarship. Aniket Didolkar would like to thank Mila for providing part of the compute resources used in this work.

## CONTRIBUTIONS

This project was initiated by AD and MS, and MS had the role of project lead. AD and MS contributed equally. AZ joined the project from the start and had critical input at all stages. AD, AZ, and MS shaped the project direction, with advise from AG, MM, YB, and GM. MS implemented most of the code, with contributions from AD. AD and MS performed the exploratory experiments. AD performed most of the final experiments and evaluations, with some experiments ran by MS. AZ performed the analysis of encoder features and created the corresponding figures (Fig. 4, Fig. A.4), AD created the model figure (Fig. 3), and MS created the remaining figures. The first draft was written by AD, AZ and MS, with AG and GM contributing to the final version.

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

# APPENDIX

## A  ADDITIONAL EXPERIMENTS

### A.1  ZERO-SHOT BENCHMARK

We show additional results complementary to the results in the main part. Figure A.1 shows benchmark results in terms of varying models, training data distribution, and training dataset size, but with the mBO metric instead of the FG-ARI metric. The results largely mirror those in Fig. 1; it can be seen that DINOSAUR generally has worse mBO than SPOT and SlotDiffusion, whereas with FG-ARI, this trend is reversed.

Figure A.3 shows the data scaling behavior of our FT-DINOSAUR method trained on different subsets of the COCO dataset, showing performance on the individual datasets in Fig. A.3a, and comparing the aggregated performance to DINOSAUR in Fig. A.3b. While DINOSAUR is better in the very-low sample regime (less than 5 000 samples), FT-DINOSAUR overall shows better scaling behavior. In particular, FT-DINOSAUR exhibits a slightly upward trending scaling curve for OOD evaluation with FG-ARI; while the effect is too weak to conclude that FT-DINOSAUR scales well with data, it would be interesting to extend this experiment to include 1–2 magnitudes more data.

### A.2  OBJECT-CENTRIC FINETUNING

**Targets from EMA teacher**   We can also frame our setup as a variant of the student-teacher framework common in self-supervised methods (Grill et al., 2020; Caron et al., 2021; Oquab et al., 2023). There, the weights of the teacher model are continuously updated from the student's weights through an exponential moving average (EMA), with a momentum parameter $\tau \in [0, 1]$ controlling the speed of adaptation. Through this lens, our approach uses $\tau = 1$, corresponding to not updating the teacher. This view suggests to use $\tau < 1$ to improve the targets throughout training.

**Table A.1**: **Analysis of targets.** $\tau$ is the momentum for teacher updates.

| $\tau$ | FG-ARI | mBO |
|---|---|---|
| 0 | 0.09 | 13.6 |
| 0.999 | 23.6 | 16.4 |
| 0.9999 | 37.8 | 21.0 |
| 1 | 48.5 | 30.7 |

In Table A.1, we analyze the effect of introducing student-teacher style EMA updates. Directly using the features of the student as the targets ($\tau = 0$) leads to collapse, as reported previously (Seitzer et al., 2023). With momentum updates, we still find a high value for $\tau$ to be necessary to stabilize training. Using fixed targets ($\tau = 1$) gives the best results. We speculate this is because there is no missing information in the auto-encoder setup, leading to a gradual loss of information.

**Training Encoder from Scratch**   Here, we build upon the approach mentioned in Section 4.1 where the model is trained with a ViT encoder initialized from scratch while still predicting DINOv2 features

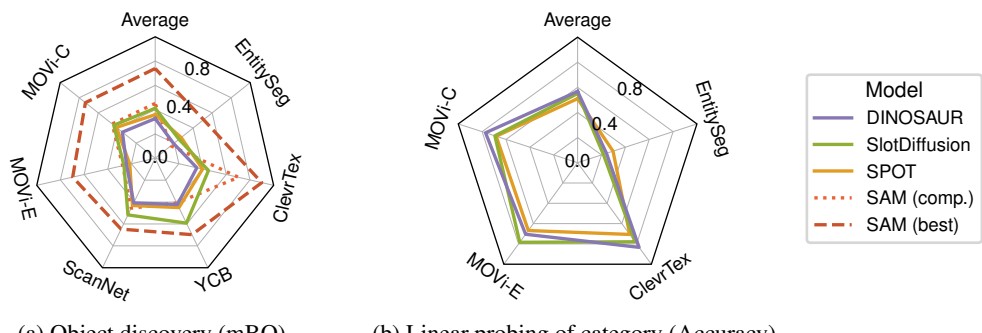

(a) Object discovery (mBO).         (b) Linear probing of category (Accuracy).

**Figure A.1**: **Evaluating zero-shot transfer of current object-centric models** trained on the COCO dataset. Corresponds to Fig. 1, but shows **(a)** mBO instead of FG-ARI and **(b)** linear probing accuracy of object category from slots.

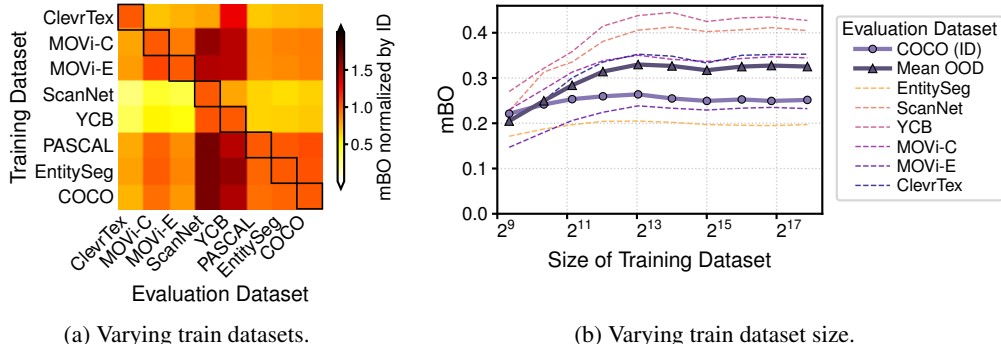

(a) Varying train datasets.

(b) Varying train dataset size.

**Figure A.2**: **Effect of training data on transfer of the DINOSAUR model.** Corresponds to Fig. 2, but shows mBO instead of FG-ARI.

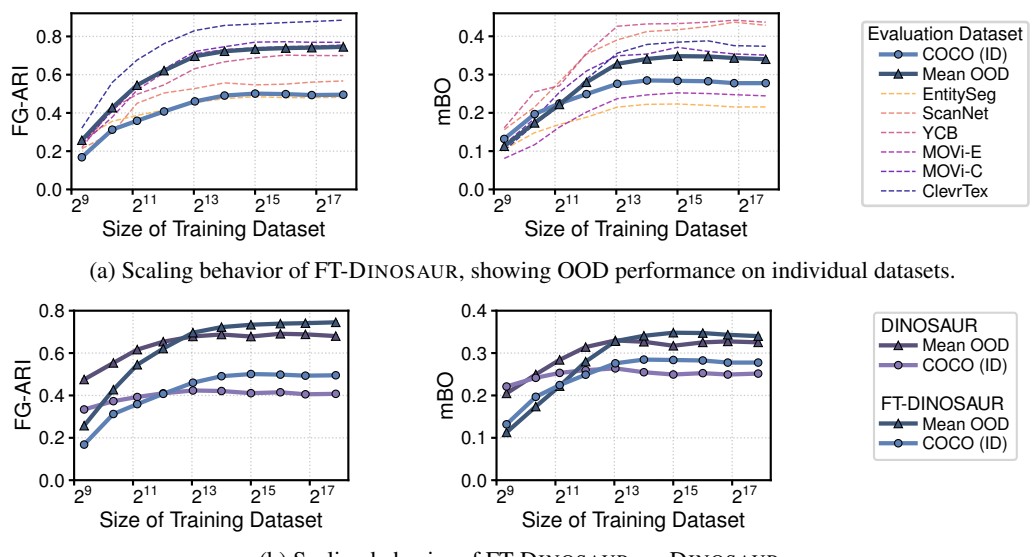

(a) Scaling behavior of FT-DINOSAUR, showing OOD performance on individual datasets.

(b) Scaling behavior of FT-DINOSAUR vs. DINOSAUR.

**Figure A.3**: **Scaling behaviour of FT-DINOSAUR on trained on differently sized subsets of COCO.** Our method uses a ViT-S/14 with DINOv2 with finetuning, but no top-k decoding and hi-res adaptation.

**Table A.2**: **Zero-shot performance without a pre-trained encoder.** Comparison of FG-ARI and mBO across multiple datasets. Scratch-DINOSAUR is identical to FT-DINOSAUR except that the ViT encoder is initialized from scratch and trained to predict DINOv2 targets. The average is computed as a weighted average, normalizing with the size of the evaluation datasets.

| | Movi-C | | Movi-E | | ScanNet | | Ycb | |
|---|---|---|---|---|---|---|---|---|
| | FG-ARI | mBO | FG-ARI | mBO | FG-ARI | mBO | FG-ARI | mBO |
| DINOSAUR | 67.0 | 34.5 | **71.1** | 24.2 | **57.4** | 40.8 | 60.2 | 42.2 |
| SlotDiffusion | 66.9 | 43.6 | 67.6 | 26.4 | 52.0 | **51.7** | 62.5 | **59.2** |
| SPOT | 63.0 | 40.8 | 47.8 | 21.5 | 48.6 | 43.2 | 52.9 | 45.1 |
| FT-DINOSAUR (ViT-S/14) | 71.3 | **44.2** | **71.1** | **29.9** | 54.8 | 48.4 | 67.4 | 54.5 |
| Scratch-DINOSAUR (ViT-S/14) | 68.3 | 31.9 | 67.9 | 23.1 | 50.5 | 36.9 | 57.3 | 35.8 |

| | ClevrTex | | EntitySeg | | COCO | | Average | |
|---|---|---|---|---|---|---|---|---|
| | FG-ARI | mBO | FG-ARI | mBO | FG-ARI | mBO | FG-ARI | mBO |
| DINOSAUR | 82.5 | 35.2 | 43.5 | 19.4 | 42.4 | 29.4 | 68.4 | 32.7 |
| SlotDiffusion | 77.0 | 45.0 | 43.7 | 25.1 | 37.3 | 31.4 | 65.8 | 41.2 |
| SPOT | 63.3 | 40.0 | 41.7 | 27.4 | 37.0 | 34.8 | 55.4 | 35.9 |
| FT-DINOSAUR (ViT-S/14) | **86.0** | **50.1** | 48.1 | 28.4 | **48.5** | **30.7** | **71.2** | **42.9** |
| Scratch-DINOSAUR (ViT-S/14) | 78.8 | 31.4 | **46.6** | **22.7** | 42.3 | 27.3 | 67.1 | 36.4 |

as targets. Note that, fully removing the pre-trained targets is not feasible on real-world data because, in that case, the representations collapse and the model fails to discover objects. In this setting, the model is trained solely on the COCO dataset, and we refer to this variant as *Scratch-DINOSAUR*.

Table A.2 presents the results of zero-shot object discovery across the benchmark. Even though the encoder was not pre-trained, Scratch-DINOSAUR achieves 42.3 FG-ARI and 27.3 mBO on COCO, and it transfers to the other datasets with performance mostly on par with DINOSAUR in FG-ARI (even outperforming certain baselines in some cases), albeit with a drop in mBO. Thus, a pre-trained encoder is *not* strictly mandatory for robust zero-shot transfer of object-centric representations, but the combination of pre-trained features with slot attention remains advantageous. In fact, adding our proposed finetuning strategy (FT-DINOSAUR) on top of a high-quality encoder yields the best performance overall.

**Zero-Shot Performance with Learnable Slot Initialization**   We investigate the impact of learnable slot initialization on zero-shot object discovery performance. Unlike our proposed approach, which uses random slot initialization, this variant employs learnable initial slots while keeping all other components unchanged.

On COCO, this model achieves 44.3 FG-ARI and 29.9 mBO, compared to 48.5 FG-ARI and 30.7 mBO achieved by our proposed method. To further evaluate its effectiveness, we test the model under zero-shot conditions on multiple datasets. We denote this variant as *FT-DINOSAUR (learnable init.)* and summarize its performance in Table A.3.

As shown in Table A.3, the learnable slot initialization approach performs competitively in FG-ARI but significantly underperforms in mBO compared to our proposed FT-DINOSAUR method. This indicates that while the learnable initialization still allows for reasonable object discovery, it produces less precise segmentation masks, leading to reduced mask quality.

Additionally, using learnable slots has a major drawback in zero-shot settings: the number of slots is fixed at training time and cannot be adjusted during inference. In contrast, our proposed random initialization approach enables the model to generalize across different slot configurations at test time, which is particularly useful when dealing with novel datasets.

Overall, these results validate the importance of using random initialization for achieving both strong object discovery and high-quality segmentation masks.

**Analysis of Finetuned Features**   In Fig. A.4, we show additional examples for visualizing the PCA on the finetuned features compared to DINOv2 features (similar to Fig. 4). Similar to the discussion

**Table A.3**: **Zero-shot performance with learnable slot initialization.** We compare FG-ARI and mBO across datasets for FT-DINOSAUR variants. The learnable initialization approach performs competitively in FG-ARI but shows significantly lower mBO, indicating weaker mask sharpness.

| | MOVI-C | | MOVI-E | | SCANNET | | YCB | |
|---|---|---|---|---|---|---|---|---|
| | FG-ARI | mBO | FG-ARI | mBO | FG-ARI | mBO | FG-ARI | mBO |
| FT-DINOSAUR (learnable init.) | 72.2 | 30.8 | 65.6 | 18.2 | 54.9 | 42.9 | 67.3 | 42.8 |
| FT-DINOSAUR (ViT-S/14) | 71.3 | **44.2** | **71.1** | **29.9** | 54.8 | 48.4 | 67.4 | 54.5 |
| FT-DINOSAUR (ViT-B/14) | **73.3** | 42.9 | 69.7 | 27.9 | **55.8** | **48.6** | **70.1** | **54.1** |

| | CLEVRTEX | | ENTITYSEG | | COCO | | Average | |
|---|---|---|---|---|---|---|---|---|
| | FG-ARI | mBO | FG-ARI | mBO | FG-ARI | mBO | FG-ARI | mBO |
| FT-DINOSAUR (learnable init.) | 81.7 | 31.8 | 47.7 | 24.2 | 44.3 | 29.9 | 68.2 | 31.3 |
| FT-DINOSAUR (ViT-S/14) | **86.0** | **50.1** | 48.1 | 28.4 | **48.5** | **30.7** | 71.2 | **42.9** |
| FT-DINOSAUR (ViT-B/14) | 83.9 | 45.9 | **49.7** | **29.0** | 46.9 | 29.5 | **71.5** | 40.8 |

in Sec. 4.2, we find that after finetuning, the encoder features are noticeably more object-centric. For example, in the first and last examples, DINOv2 features show a part-based split of the shown persons in the dominant PCA components; the finetuned features highlight the whole persons better. In the second example, DINOv2 features group semantic instances (human) together in the dominant components; after finetuning, the features clearly split the persons. However, note that is not necessary that the features highlight the instances in the dominant components to derived an instance-based grouping; in all examples, the masks discovered by DINOSAUR (last column) feature a correct instance split (while also splitting further into parts in the last two examples). This may be because the necessary information for the correct split is contained in the less dominant components of the features (e.g. in PCA dimensions 4–6). However, we conjecture that the finetuned features simplify the grouping task for slot attention, leading to better and more consistent object discovery.

## A.3 EVALUATION

We include the following additional results for the evaluation of FT-DINOSAUR conducted in Sec. 5 of the main paper:

- **Finetuning In-Distribution (Sec. 5.1)**: we show the numeric values corresponding to Fig. 5 in the main part in Table A.4. This table also shows the results for the in-distribution vs. zero-shot comparison in Fig. 6.

- **Extended Comparison To Prior Work on Real-World Object-Centric Learning (Sec. 5.2)**: we conduct an extended comparison to prior work for real-world object-centric learning on the COCO dataset in Table A.5.

- **Zero-Shot Evaluation (Sec. 5.3)**: for object discovery, we show the full results all datasets of the zero-shot benchmark, corresponding to Fig. 1a and Fig. 7 in the main part in Table A.6. For downstream property prediction, we show the per-dataset results in Table A.7, corresponding to Fig. 1b in the main part.

## B MODEL LIMITATIONS

Even though FT-DINOSAUR brings large improvements over DINOSAUR, the model still exhibits problems with certain types of scenes. In Fig. B.5, we show several examples of such failure cases, grouped into modes of failure. Two typical categories of failure are the overgrouping of semantically-related objects (Fig. B.5a), and the split of objects into parts (Fig. B.5b). Both problems are primarily caused by the model using the wrong number of slots. But note that even having access to the "correct" number of slots per image can not resolve all problems, as the model may still allocate the slots in undesirable ways. Consider the last example in Fig. B.5a: here, the model could correctly split the two persons into individual slots if the motorbike is grouped as one object instead of as parts. A third category of failure broadly stems from difficult or unusual images (Fig. B.5c): for example, grouping

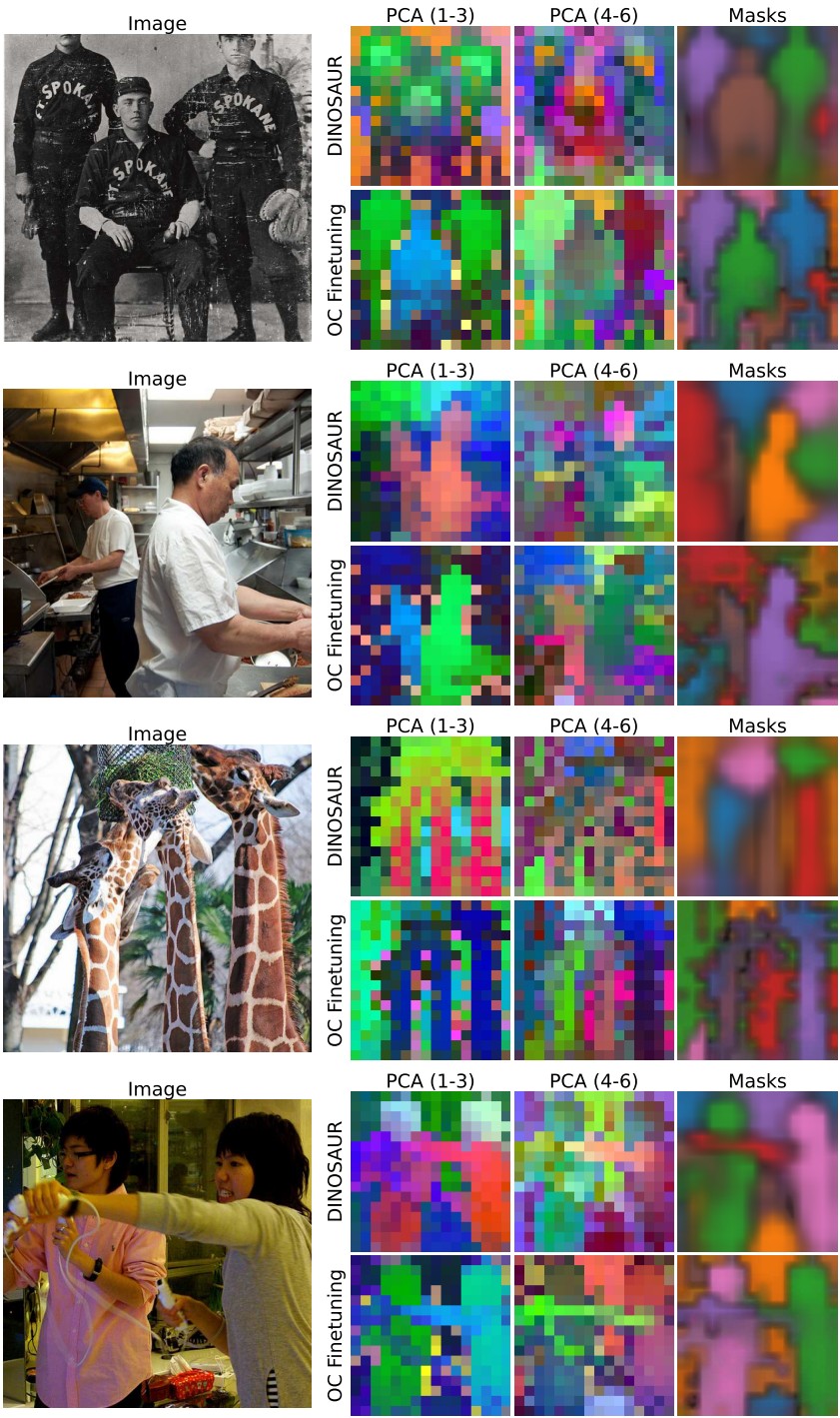

**Figure A.4**: **Visualization of encoder features** in DINOSAUR (frozen DINOv2 features) and for encoder features adapted with object-centric finetuning, similar to Fig. 4 in the main paper. The second column shows 1st to 3rd PCA components, and the third column shows 4th to 6th PCA components grouped in one image by using different RGB channels. The last column shows object discovery masks by each method.

**Table A.4**: **Evaluation of adding _finetuning_ to DINOSAUR** when training _in-distribution_, using a ViT-S/14 DINOv2 backbone. Finetuning shows strong performance improvements on all eight datasets. We also show transfer when finetuning on COCO, which performs comparable or better to training in-distribution on 5 out of 7 datasets. Results corresponding to experiment in Fig. 5 in the main paper.

| | MOVI-C | | MOVI-E | | SCANNET | | YCB | |
|---|---|---|---|---|---|---|---|---|
| | FG-ARI | mBO | FG-ARI | mBO | FG-ARI | mBO | FG-ARI | mBO |
| DINOSAUR | 63.1 | 33.3 | 74.0 | 25.5 | 52.8 | 38.8 | 67.5 | 28.9 |
| +Finetuning | 71.9 | 36.8 | 82.0 | 28.1 | 63.8 | 43.8 | 71.1 | 27.5 |
| Zero-shot (COCO) | 76.4 | 34.9 | 75.1 | 24.5 | 55.4 | 42.8 | 69.9 | 44.4 |

| | CLEVRTEX | | PASCAL VOC | | ENTITYSEG | | COCO | |
|---|---|---|---|---|---|---|---|---|
| | FG-ARI | mBO | FG-ARI | mBO | FG-ARI | mBO | FG-ARI | mBO |
| DINOSAUR | 91.3 | 40.1 | 26.2 | 40.1 | 43.0 | 19.6 | 42.4 | 29.4 |
| +Finetuning | 95.8 | 40.1 | 29.7 | 40.8 | 45.7 | 20.9 | 48.5 | 30.7 |
| Zero-shot (COCO) | 87.5 | 38.2 | 31.7 | 40.4 | 47.7 | 21.8 | 48.5 | 30.7 |

**Table A.5**: **Extended comparison of FT-DINOSAUR to prior work on the COCO dataset**, corresponding to Table 2 in the main paper. For our proposed approach, we average results across 5 seeds for FT-DINOSAUR, ViT-S/14 and across 3 seeds for FT-DINOSAUR, ViT-B/14. Results marked with † are from evaluating official checkpoints; results marked with * are taken from the respective papers. Supervised models (SAM) colored in gray.

| Model | FG-ARI | mBO | P-ARI | PQ |
|---|---|---|---|---|
| Slot Attention (Locatello et al., 2020), (Wu et al., 2023b)* | 21.4 | 17.2 | | |
| SLATE (Singh et al., 2022a), (Wu et al., 2023b)* | 32.5 | 29.1 | | |
| DINOSAUR (MLP Dec.) (Seitzer et al., 2023) | 40.5 | 27.7 | 37.1 | 14.4 |
| DINOSAUR (TF. Dec.) (Seitzer et al., 2023)* | 34.1 | 31.6 | | |
| Stable-LSD (Jiang et al., 2023)* | 35.0 | 30.4 | | |
| SlotDiffusion (Wu et al., 2023b)† | 37.3 | 31.4 | 47.6 | 21.0 |
| SPOT (Kakogeorgiou et al., 2024)† | 37.0 | 34.8 | **52.4** | 21.3 |
| FT-DINOSAUR, ViT-S/14 | 46.6 | 35.6 | 49.7 | 23.5 |
| FT-DINOSAUR, ViT-B/14 | **48.8** | **36.3** | 49.4 | **23.9** |
| SAM (comp.) (Kirillov et al., 2023)† | 12.1 | 19.0 | 10.8 | 9.4 |
| SAM (best.) (Kirillov et al., 2023)† | 44.9 | 56.9 | 54.4 | 10.9 |

**Table A.6**: **Per-dataset zero-shot performance**, corresponding to Fig. 7 in the main paper. All unsupervised object-centric methods (DINOSAUR, SlotDiffusion, SPOT, FT-DINOSAUR) are trained on the COCO dataset. Furthermore, we compare with the supervised Segment Anything model (SAM). Average is computed as a weighted average normalizing with the size of the evaluation datasets (cf. Table E.9), and only includes the zero-shot datasets (first 6 columns). For completeness, we also include the performance when transferring from COCO to PASCAL VOC in the last column. Results marked † are from evaluating official checkpoints. Supervised models (SAM) are colored in gray.

| | MOVI-C | | MOVI-E | | SCANNET | | YCB | |
|---|---|---|---|---|---|---|---|---|
| | FG-ARI | mBO | FG-ARI | mBO | FG-ARI | mBO | FG-ARI | mBO |
| DINOSAUR | 67.0 | 34.5 | **71.1** | 24.2 | **57.4** | 40.8 | 60.2 | 42.2 |
| SlotDiffusion† | 66.9 | 43.6 | 67.6 | 26.4 | 52.0 | **51.7** | 62.5 | **59.2** |
| SPOT † | 63.0 | 40.8 | 47.8 | 21.5 | 48.6 | 43.2 | 52.9 | 45.1 |
| FT-DINOSAUR (ViT-S/14) | 71.3 | **44.2** | **71.1** | **29.9** | 54.8 | 48.4 | 67.4 | 54.5 |
| FT-DINOSAUR (ViT-B/14) | **73.3** | 42.9 | 69.7 | 27.9 | 55.8 | 48.6 | **70.1** | 54.1 |
| SAM (comp.)† | 57.6 | 45.3 | 38.5 | 27.4 | 45.8 | 45.5 | 46.9 | 40.9 |
| SAM (best.)† | 79.7 | 73.5 | 84.7 | 69.7 | 62.2 | 64.7 | 69.4 | 69.8 |

| | CLEVRTEX | | ENTITYSEG | | Average | | PASCAL VOC | |
|---|---|---|---|---|---|---|---|---|
| | FG-ARI | mBO | FG-ARI | mBO | FG-ARI | mBO | FG-ARI | mBO |
| DINOSAUR | 82.5 | 35.2 | 43.5 | 19.4 | 68.4 | 32.7 | 24.0 | 37.2 |
| SlotDiffusion† | 77.0 | 45.0 | 43.7 | 25.1 | 65.8 | 41.2 | 21.1 | 42.0 |
| SPOT † | 63.3 | 40.0 | 41.7 | 27.4 | 55.4 | 35.9 | 21.2 | **50.6** |
| FT-DINOSAUR (ViT-S/14) | **86.0** | **50.1** | 48.1 | 28.4 | 71.2 | **42.9** | 24.0 | 37.6 |
| FT-DINOSAUR (ViT-B/14) | 83.9 | 45.9 | **49.7** | **29.0** | **71.3** | 41.1 | **25.9** | 37.8 |
| SAM (comp.)† | 82.9 | 70.3 | 25.9 | 16.5 | 55.2 | 45.7 | 31.0 | 51.5 |
| SAM (best.)† | 94.0 | 90.0 | 53.4 | 51.0 | 79.5 | 73.2 | 31.1 | 64.2 |

**Table A.7**: **Per-dataset downstream property linear prediction performance.** We report in-distribution property prediction performance for each dataset (DINOSAUR (ID)) and zero-shot performance of models trained on the COCO dataset (DINOSAUR, SlotDiffusion, SPOT, FT-DINOSAUR, FT-DINOSAUR + Hi-Res). Results marked † are from evaluating official checkpoints.

| | MOVI-C | | MOVI-E | | YCB | |
|---|---|---|---|---|---|---|
| | Acc (%) | $R^2$ | Acc (%) | $R^2$ | Acc (%) | $R^2$ |
| DINOSAUR (ID) | 70.7 | 0.69 | 64.3 | 0.44 | – | 0.54 |
| DINOSAUR | 77.1 | 0.69 | 70.4 | 0.45 | – | 0.77 |
| SlotDiffusion† | 69.3 | 0.58 | 78.4 | 0.29 | – | 0.76 |
| SPOT † | 68.2 | 0.55 | 66.8 | 0.35 | – | 0.76 |
| FT-DINOSAUR (ViT-S/14) | 75.9 | 0.65 | 67.4 | 45.5 | – | 0.83 |
| FT-DINOSAUR (ViT-S/14) + Hi-Res | 74.6 | 0.65 | 69.3 | 0.47 | – | 0.83 |

| | CLEVRTEX | | ENTITYSEG | | SCANNET | |
|---|---|---|---|---|---|---|
| | Acc (%) | $R^2$ | Acc (%) | $R^2$ | Acc (%) | $R^2$ |
| DINOSAUR (ID) | 90.7 | 0.90 | 25.9 | 0.53 | – | – |
| DINOSAUR | 83.4 | 0.76 | 24.1 | 0.54 | – | 0.80 |
| SlotDiffusion† | 77.8 | 0.67 | 21.5 | 0.36 | – | 0.82 |
| SPOT † | 70.7 | 0.55 | 29.7 | 0.42 | – | 0.82 |
| FT-DINOSAUR (ViT-S/14) | 77.8 | 0.68 | 18.1 | 0.45 | – | 0.80 |
| FT-DINOSAUR (ViT-S/14) + Hi-Res | 79.9 | 0.70 | 19.4 | 0.51 | – | 0.79 |

tiny objects together with the background (cars on bridge); incorrect 3D inference due to unusual camera perspective (rail and light post); sub-optimal decompositions in OOD scenes (grass stalk in forest, sand patterns).

*How could the failure modes regarding overgrouping and oversplitting be resolved?* First, like all slot attention/DINOSAUR-based methods, FT-DINOSAUR decomposes the scene into a fixed number of regions/objects. However, especially on real-world images, the number of objects varies significantly from image to image. Therefore, it is important to develop methods that infer a suitable number of objects for an image; however, further innovations are needed to deal with the slot allocation problem we have alluded to before. Second, unsupervised scene decomposition is inherently an ill-defined task on real-world data as scenes can be split in numerous ways (cf. Figs. B.5a and B.5b). Thus, predicting only a single set of masks might ultimately be insufficient. Instead, it may be beneficial to model the full *part-whole hierarchy*, producing various decompositions of different granularity. Such models could further allow *control* over the level of granularity through external conditioning variables or text. However, the examples in Figs. B.5a and B.5b also demonstrate the limitations of current evaluation techniques. Arguably, these are not failures of the model, but are treated as such by the evaluation metrics. This is because current datasets have annotations that prescribe a single ground truth labeling for each image. Instead, datasets should be annotated with multi-level labelings, e.g. by including parts of objects, or further splitting the background into specific elements (e.g. splitting the background "tree" class into particular trees). To evaluate methods that model the full part-whole hierarchy, such annotations even become a necessity.

## C  METHOD DETAILS

### C.1  DINOSAUR

DINOSAUR (Seitzer et al., 2023) proposes to use pretrained encoders for unsupervised object discovery. The image is processed using a pretrained DINO encoder (Caron et al., 2021) into a set of feature vectors $\{\boldsymbol{f}_1, \boldsymbol{f}_2, \ldots \boldsymbol{f}_N\}$. The slot attention module (see App. C.2) is applied to the outputs of the pretrained encoder to produce a set of slot vectors $\{\boldsymbol{s}_1, \boldsymbol{s}_2, \ldots, \boldsymbol{s}_M\}$. The slots are decoded using a broadcast MLP decoder (Watters et al., 2019) to produce a set of feature vectors $\hat{\boldsymbol{f}}_1, \hat{\boldsymbol{f}}_2, \ldots \hat{\boldsymbol{f}}_N$.

The MLP decoder decodes each slot $\boldsymbol{s}_k$ separately to output a mask logit $\hat{m}_{ik}$ and feature values $\hat{f}_{ik}$. Here, $i \in \{1, \ldots N\}$ indexes the features and $k \in \{1, \ldots, K\}$ indexes the slots. The mask logits are converted to probabilities $m_{ik}$ by taking their softmax across all slots. Here, $m_{ik}$ denotes the probability that slot $\boldsymbol{s}_k$ produces feature $\boldsymbol{f}_i$ and $\hat{f}_{ik}$ denotes the feature value assigned to feature $\boldsymbol{f}_i$ by slot $\boldsymbol{s}_k$. The final value for each feature is obtained by taking a weighted sum across all slots: $\hat{\boldsymbol{f}}_i = \sum_{k=1}^{K} m_{ik} \hat{f}_{ik}$. The training objective is the feature-reconstruction objective: $\mathcal{L} = \|\boldsymbol{f} - \hat{\boldsymbol{f}}\|^2$.

Note that the mask $\boldsymbol{m}_k \in \mathbb{R}^N$ is the mask obtained for a given slot. This mask is resized to the original image size and overlayed over the image to produce the visualizations shown in Table 1.

### C.2  SLOT ATTENTION

Slot attention (Locatello et al., 2020) is a differentiable clustering procedure that operates on a set of visual features (usually output by a CNN) and outputs a set of slot vectors $\{\boldsymbol{z}_1, \boldsymbol{z}_2, \ldots \boldsymbol{z}_K\}$ where each slot represents an object. The differentiable clustering procedure is iterative in nature. We now describe the procedure in detail.

Consider that we have a set of features $\{\boldsymbol{f}_1, \ldots \boldsymbol{f}_N\}$, where $\boldsymbol{f}_i \in \mathbb{R}^D$, and a set of slots $\{\boldsymbol{z}_1, \ldots, \boldsymbol{z}_K\}$, where $\boldsymbol{z}_i \in \mathbb{R}^D$. The features $\{\boldsymbol{f}_1, \ldots \boldsymbol{f}_N\}$ are output by a CNN encoder given input image $\boldsymbol{x}$ and the slots are sampled from a Gaussian distribution. The attention mechanism computes the assignment of features to slots. Ideally, the set of features belonging to a particular object will map to a single slot and each slot will bind to a distinct object. To induce this behavior, the attention mechanism implements a competition between the slots to represent parts of the feature space.

First, following the general process of query-key-value (QKV) attention (Vaswani, 2017), the slots are projected to queries $\boldsymbol{q}_i = \text{Linear}_q(\boldsymbol{z}_i)$ and the features are projected to keys $\boldsymbol{k}_i = \text{Linear}_k(\boldsymbol{f}_i)$ and values $v_i = \text{Linear}_v(\boldsymbol{f}_i)$. This results in queries $\boldsymbol{q} \in \mathbb{R}^{K \times D}$, keys $\boldsymbol{k} \in \mathbb{R}^{N \times D}$, and values

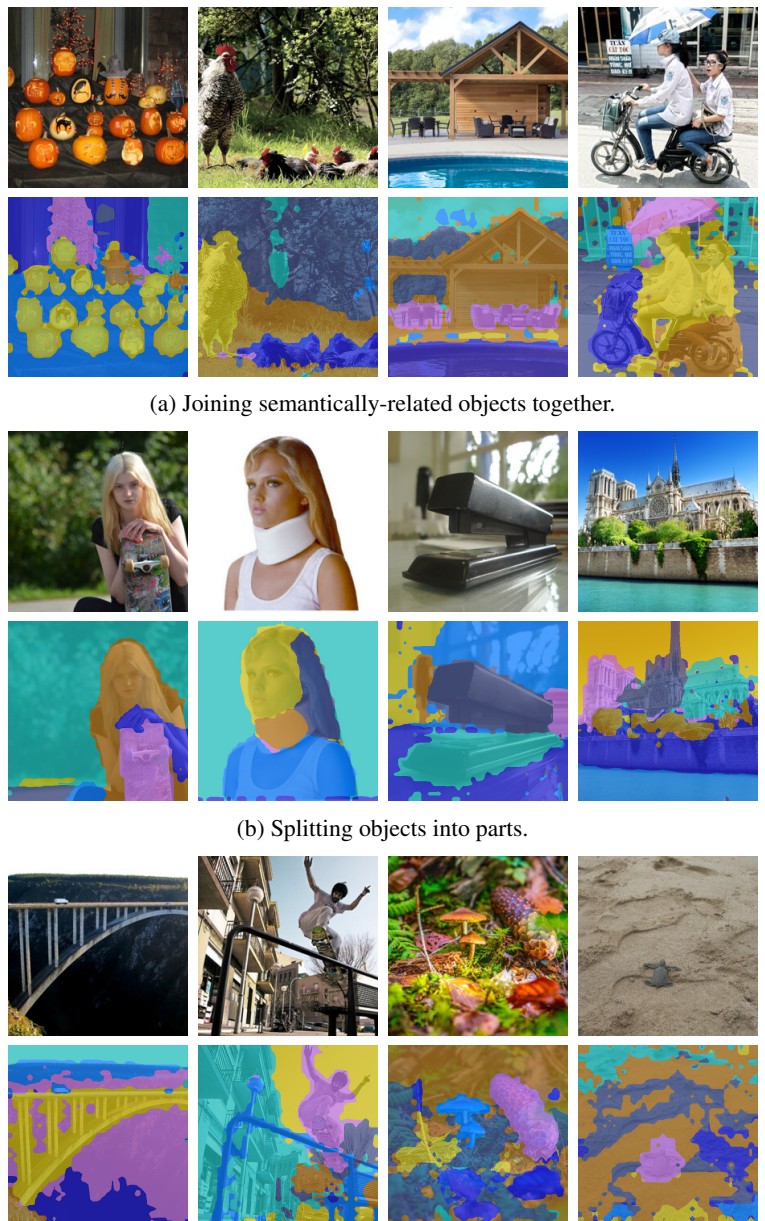

(a) Joining semantically-related objects together.

(b) Splitting objects into parts.

(c) Complex or unusual images: (1) tiny objects, (2) incorrect 3D inference, (3, 4) OOD scenes.

**Figure B.5**: **Failure modes of FT-DINOSAUR.** We show typical failure cases grouped into three categories: (a) joining semantically related objects into a single object; (b) splitting objects into parts; and (c) incorrect decomposition of complex or unusual scenes. Note that the model's decompositions in (a) and (b) arguably are correct but do not correspond to the labeling prescribed by the ground truth annotations; without knowledge of the intended downstream task, the "correct" grouping is ambiguous. We use the model from Table 2; it uses a ViT-B/14 encoder with hi-res adaptation and is trained on the COCO dataset. All images show zero-shot predictions on the ENTITYSEG dataset.

$v \in \mathbb{R}^{N \times D}$. The dot-product attention then computes the affinities between the queries and the keys: $\text{attn} = q \times k^T$, where $\text{attn} \in \mathbb{R}^{N \times K}$. To induce a competition between the slots to represent the features, they take a softmax across the *queries* as opposed to taking it across the *keys* which is what is usually done in QKV attention. Therefore, the final attention values become $\text{attn} = \text{softmax}(\frac{q \times k^T}{\sqrt{D}}, \text{axis} = q)$, where $\sqrt{D}$ is the softmax temperature. The final attention outputs $\hat{z}$ are obtained by taking a weighted mean of the values $\hat{z} = W \cdot v$, where $W_{ij} = \frac{\text{attn}_{ij}}{\sum_{n=1}^{N} \text{attn}_{in}}$. Therefore, the attention computation used in each iteration of slot attention can be implemented as follows:

$$\text{attn} = \text{softmax}\left(\frac{q(z) \cdot k(f)^T}{\sqrt{D}}, \text{axis} = \text{query}\right), \tag{1}$$

$$\text{attn} = \frac{\text{attn}}{\text{Sum}(\text{attn}, \text{axis} = \text{keys})}, \tag{2}$$

$$\hat{z} = \text{attn} \cdot v(f). \tag{3}$$

The authors find it useful from a stability viewpoint to process the inputs $f$ and slots $z$ using layer normalization. The output of the attention mechanism is used to update the current estimate of the slots in an iterative procedure which will describe next.

**Iterative Procedure**    The iterative updates to the slots are carried out using a GRU and an MLP. The slots $z_1, \ldots z_K$ are first initialized as the hidden states of a GRU. Given the updates from the attention mechanism, the slots are updated as follows:

$$z_i = \text{GRU}(\text{input} = \hat{z}_i, \text{state} = z_i) \quad z_i \qquad = \text{MLP}(\text{LayerNorm}(z_i)) + z_i. \tag{4}$$

These updates happen for a fixed number of iterations $T$ which is set as a hyperparameter.

This whole process can be viewed as soft k-means clustering. The slots are analogous to cluster centers and each attention computation computes the assignments of features to slots. Next, the slots (cluster centers) are updated with the information from the features assigned to each slot.

## C.3    Top-K Decoding

We first recap the MLP decoder from DINOSAUR Seitzer et al. (2023). For $N$ patches and $K$ slots, the MLP decoder produces a reconstruction $\hat{y} \in \mathbb{R}^{N \times K \times D}$, as well as an alpha mask $\alpha \in \mathbb{R}^{N \times K}$ that shows how active each slot is at each patch. The final reconstruction $y \in \mathbb{R}^{N \times D}$ is then given by taking a weighted average over the slots, that is, the reconstruction $y_i$ for patch $i$ is given by

$$y_i = \sum_{\kappa=1}^{K} \hat{y}_{i,\kappa} \odot m_{i,\kappa}, \qquad m_{i,\kappa} = \left(\text{softmax}_{j} \, \alpha_{i,j}\right)_{\kappa}. \tag{5}$$

With **top-k decoding**, we only take the $k \in \mathcal{K}_i$ most active slots into account for each patch $i$, as determined by the slot attention mask $a \in [0, 1]^{N \times K}$:

$$y_i = \sum_{\kappa \in \mathcal{K}_i} \hat{y}_{i,\kappa} \odot m_{i,\kappa}, \qquad m_{i,\kappa} = \left(\text{softmax}_{j \in \mathcal{K}^i} \, \alpha_{i,j}\right)_{\kappa}, \qquad \mathcal{K}_i = \text{topk}\left(a_i, k\right), \tag{6}$$

where $\text{topk}(x, k) = \arg\max_{I \subseteq \{1, \ldots, n\}: |I| = k} \sum_{i \in I} x_i$ is the function that selects the indices of the $k$ highest values of the vector $x \in \mathbb{R}^n$. In practice, we can efficiently implement the decoding step by first broadcasting slots to patches and adding the positional encoding, then packing the top-k slots for each position together using a gather operation, directly resulting in reconstructions $\hat{y} \in \mathbb{R}^{N \times k}$ and alpha masks $\alpha \in \mathbb{R}^{N \times k}$.

## C.4    Improved Hyperparameters

As discussed in Sec. 4.1 in the main paper, we found an improved set of hyperparameters that work well for finetuning the pre-trained ViT encoder. We split these into general hyperparameters

(G-HPs), affecting all modules of the model, and encoder hyperparameters (E-HPs), only affecting the finetuning of the encoder (see also Table C.8). We ablate the effect of these groups of hyperparameters in Table 1 in the main paper.

The general hyperparameter changes are as follows:

- Increasing *batch size* from 64 to 128.

- Decreasing *base learning rate* from 0.0004 to 0.0003.

- Switching from an exponential decay *learning rate schedule* to a cosine schedule.

- Lowering *gradient clipping* from 1.0 to 0.1.

The hyperparameter for encoder finetuning are as follows:

- Lowering the *base learning rate* for the encoder by a factor of 0.5 from 0.0003 to 0.00015.

- Introducing *blockwise learning rate decay* with a decay rate of 0.85.

- Adding *weight decay* of 0.01 to the encoder parameters in conjunction with the AdamW optimizer.

Note that these changes resulted from a joined hyperparameter search over all individual hyperparameters and it is highly likely that (1) not all of these parameters changes are necessary, and (2) an even better set of hyperparameters can be found.

## D  METHODS & HYPERPARAMETERS

**DINOSAUR (Seitzer et al., 2023)**  DINOSAUR introduced the idea of applying slot attention on a pre-trained encoder and training the model by reconstructing the features of this pre-trained encoder. This also forms the base of our proposed approach. In DINOSAUR, the encoder is kept fixed while the slot attention and the decoder modules are trainable. While the original paper considers two kinds of decoders — (1) Transformer Decoder and (2) MLP decoder — in this work we mainly compare against DINOSAUR with the MLP decoder. We consider two variants of DINOSAUR, using DINO (Caron et al., 2021) and DINOv2 (Oquab et al., 2023) pre-trained backbones respectively. We list the hyperparameters used for DINOSAUR in Table C.8 (first column).

**FT-DINOSAUR**  Our method is implemented upon DINOSAUR and thus shares low-level implementation details. In Table C.8, we list the hyperparameters for the following models mentioned in Sec. 4 and listed in Table 1: (1) DINOSAUR + Training from Random Init., (2) DINOSAUR + FT w/G-HP's, (3) DINOSAUR + FT w/G-HP's & E-HP's, (4) DINOSAUR + FT, + Top-k, + High-Res. Finetuning. While the models listed in Table Table C.8 all use DINOv2 with the ViT-S/14 backbone, the same hyperparamters are applicable for models using ViT-B/16 and ViT-B/14 backbones as well. For training our model, we use a single A100 GPU per run. Each training run of the proposed finetuning approach requires 2–3 days of training.

**SlotDiffusion (Wu et al., 2023b)**  SlotDiffusion utilizes a latent diffusion model as the decoder. The specific variant of the SlotDiffusion model which we consider here is the one which uses a pretrained DINO encoder (ViT-B/16) to encode the images similar to DINOSAUR. We use the pre-trained checkpoint released by the authors[2] for all the comparisons in this work.

**SPOT (Kakogeorgiou et al., 2024)**  SPOT uses a two-stage training procedure. In the first stage, a DINOSAUR model is trained similar to Seitzer et al. (2023). In the second stage, a student-teacher setup is employed, where the model trained model from the first stage acts as a teacher and the student is a new model. During this stage, the model is trained with two objectives: (1) a feature reconstruction loss, where the targets come from the teacher, and (2) a attention distillation loss, where the teachers attention masks from slot attention are distilled into the student. Moreover, SPOT uses a Transformer encoder as opposed to an MLP decoder. Similar to SlotDiffusion, for SPOT,

---

[2]https://github.com/Wuziyi616/SlotDiffusion

**Table C.8**: **Hyperparameters for the DINOSAUR and FT-DINOSAUR models** displayed in Table 1. The second column (DINOSAUR +Enc. Train (random init.)) also lists hyperparameters for training with random encoder initialization, as discussed in Sec. 4.1. Results in Fig. 5 use the settings in the fourth column (DINOSAUR +FT w/G-HP's w/E-HP's). Results in Table 2 and Fig. 7 use the settings in the last column, but with a ViT-B/14 encoder. See also App. C.4 for a concise description of the improved hyperparameters for finetuning (G-HP's and E-HP's).

| **Models** | | **DINOSAUR** | **DINOSAUR +Enc. Train (random init.)** | **DINOSAUR +FT. w/G-HP's** | **DINOSAUR +FT w/G-HP's w/E-HP's** | **DINOSAUR +FT +Top-k +High-Res.** |
|---|---|---|---|---|---|---|
| Training Steps | | 300k | 300k | 300k | 300k | 10k |
| Batch Size | | 64 | 128 | 128 | 128 | 64 |
| Image/Crop Size | | 224 | 224 | 224 | 224 | 518 |
| Cropping Strategy | | Random | Random | Random | Random | Random |
| Augmentations | | – | – | – | – | – |
| Image Tokens | | 784/256 | 256 | 256 | 256 | 1369 |
| LR | LR Warmup Steps | 10000 | 10000 | 10000 | 10000 | 333 |
| | Base LR $\left(\text{Total LR} = \text{Base LR} \cdot \sqrt{\left(\frac{\text{Batch Size}}{64}\right)}\right)$ | 0.0004 | 0.0003 | 0.0003 | 0.0003 | 0.0001 |
| | Exp. Decay Half-Life | 100k | 100k | 100k | 100k | 10k |
| | Schedule | Exponential | Cosine | Cosine | Cosine | Cosine |
| Encoder LR | Blockwise LR | ✗ | ✓ | ✗ | ✓ | ✓ |
| | LR Factor $\left(\text{Total LR} = \frac{\text{Base LR}}{\text{Encoder LR Factor}} \cdot \sqrt{\left(\frac{\text{Batch Size}}{64}\right)}\right)$ | ✗ | 0.5 | ✗ | 0.5 | 0.5 |
| | Layerwise LR Decay Factor ($\eta$) $\left(\text{LR}_{\text{layer } l} = \eta \cdot \text{LR}_{\text{layer } l+1}\right)$ | ✗ | 0.85 | ✗ | 0.85 | 0.85 |
| | Weight Decay | ✗ | 0.01 | ✗ | 0.01 | 0.01 |
| Encoder | Type | ViT-B/16 / ViT-S/14 | ViT-S/14 | ViT-S/14 | ViT-S/14 | ViT-S/14 |
| | Pre-training | DINO/DINOv2 | – | DINOv2 | DINOv2 | DINOv2 |
| | Patch Size | 16/14 | 14 | 14 | 14 | 14 |
| | Feature Dim. $D_{\text{feat}}$ | 768/384 | 384 | 384 | 384 | 384 |
| | Gradient Norm Clipping | 1.0 | 0.1 | 0.1 | 0.1 | 0.1 |
| Target Encoder | Type | ViT-B/16 / ViT-S/14 | ViT-S/14 | ViT-S/14 | ViT-S/14 | ViT-S/14 |
| | Pre-training | DINO/DINOv2 | DINOv2 | DINOv2 | DINOv2 | DINOv2 |
| Slot Attention | Slots | 7 | 7 | 7 | 7 | 7 |
| | Iterations | 3 | 3 | 3 | 3 | 3 |
| | Slot Dim. $D_{\text{slots}}$ | 256 | 256 | 256 | 256 | 256 |
| | MLP Hidden Dim. | 1024 | 1024 | 1024 | 1024 | 1024 |
| Decoder | Type | MLP | MLP | MLP | MLP | MLP |
| | Layers | 4 | 4 | 4 | 4 | 4 |
| | MLP Hidden Dim. | 2048 | 2048 | 2048 | 2048 | 2048 |
| | Top-k | ✗ | ✗ | ✗ | ✗ | 3 |

**Table E.9**: **Number of images per dataset and the number of slots** used for training and evaluating on each dataset.

| Dataset | Num. Images | Num. Slots |
|---|---|---|
| COCO 2017 train | 118 287 | 7 |
| COCO 2017 validation | 5 000 | 7 |
| ENTITYSEG train | 31 789 | 7 |
| ENTITYSEG validation | 1 498 | 7 |
| PASCAL VOC 2012 train | 10 582 | 7 |
| PASCAL VOC 2012 validation | 1 449 | 7 |
| MOVI-C train | 87 633 | 11 |
| MOVI-C validation | 4 200 | 11 |
| MOVI-E train | 87 633 | 11 |
| MOVI-E validation | 4 176 | 11 |
| SCANNET train | 10 000 | 6 |
| SCANNET validation | 2 000 | 6 |
| YCB train | 10 000 | 6 |
| YCB validation | 2 000 | 6 |
| CLEVRTEX train | 40 000 | 11 |
| CLEVRTEX validation | 5 000 | 11 |

also we use a pre-trained checkpoint released by the authors[3] for the evaluations in this work. The pre-trained checkpoint uses a ViT-B/16 encoder initialized with DINO weights.

**Segment Anything (Kirillov et al., 2023)**   The Segment Anything model (SAM) is a large foundation model for object detection and segmentation trained supervised. It has three stages of training: (1) a manual stage, where the model is trained using 120k images annotated with 4.3M masks obtained from human labelers; (2) a semi-automatic stage, where the model is trained on 180k annotated with 5.9M masks partly annotated by human labelers and partly annotated by itself; and (3) a fully automatic stage, where the model is trained on 11M images with 1.1B masks annotated by the model itself. We consider 2 variants of SAM: *comp.* (Comparable) and *best*. Note that SAM includes an IoU prediction MLP which outputs an estimated IoU for each predicted mask. For the *comp.* variant, we use the ViT-Base model considering the top $K$ masks by predicted IoU, where the value of $K$ is based on the optimal number of objects for each dataset as listed in Table E.9. For the *best* variant, we use the ViT-Huge model keeping all masks above a IoU threshold $\tau$. We evaluated values for $\tau \in \{0.9, 0.95, 0.99\}$ and found that $\tau = 0.9$ works best across all datasets.

For inference, we use a single A100 GPU for each of the baselines and the proposed approach.

## E   DATASETS

This section gives a detailed description of the datasets we use in this work. See also Table E.9 for an overview over the number of images per dataset.

Note that current object-centric models are sensitive to the *number of objects*. As a concession to that, we evaluate the models with the number-of-slots parameter matching the expected complexity of the target dataset (mostly following prior work, see Table E.9). We leave it to future work to remove this limitation of the models.

**COCO (Lin et al., 2014)**   This dataset contains complex images containing real-world objects in their natural context. For training, we use the COCO 2017 dataset which consists of 118 287 images. For evaluation, we use 5 000 images from the validation sets. Similar to Seitzer et al. (2023), we use instance masks to evaluate object discovery. Additionally, we also add the task of panoptic segmentation to our evaluation suite for the COCO dataset, using the panoptic labeling provided by Kirillov et al. (2019). Panoptic segmentation combines the task of instance segmentation, which

---

[3] https://github.com/gkakogeorgiou/spot

requires the model to segment each object/foreground/thing instance, and semantic segmentation, which requires the model to segement each background/stuff class. The metrics we use for measuring panoptic segmentation are panoptic ARI and panoptic quality (see App. F). Following Seitzer et al. (2023), we evaluate square center crops, where the input images are resized to $224 \times 224$ pixels, and the targets masks are resized to $320 \times 320$ pixels.

**ENTITYSEG (Lu et al., 2023)**  This dataset consists of complex real world images spanning a diverse range of entities. In contrast to COCO, ENTITYSEG is an open-world dataset and does not have a pre-defined set of object classes. It consists of a large number of high-resolution images (71.25% and 86.23% of the images are of high resolution with at least 2 000px for the width and 1 000px for the height). Each image is annotated with high-quality fine-grained mask annotations. The version of the dataset utilized in this work consists of 31 789 images for training and 1 498 images for evaluation. We evaluate the instance segmentation masks for object discovery. As in COCO, we evaluate square center crops, where the input images are resized to $224 \times 224$ pixels, and the targets masks are resized to $320 \times 320$ pixels.

**PASCAL VOC (Everingham et al., 2012)**  Similar to Seitzer et al. (2023), we use the "trainaug" variant of the PASCAL VOC dataset for training. It consists of a total of 10 582 images for training, where 1 464 are from the segmentation train set and 9 118 are from the SBD dataset (Hariharan et al., 2011). For evaluating object discovery, we use the official instance segmentation validation split with 1 449 images. Following Seitzer et al. (2023), we evaluate square center crops, where the input images are resized to $224 \times 224$ pixels, and the targets masks are resized to $320 \times 320$ pixels.

**MOVi-C and MOVi-E (Greff et al., 2022)**  The MOVi datasets are synthetically generated video datasets consisting of multiple objects per video. Each video is generated by placing 3D scanned objects on real-world backgrounds. MOVi-C contains up to 11 objects per video and MOVI-E contains up to 23 objects per video. Additionally, MOVi-E also features the camera moving in random directions. For our case, we treat these datasets as image datasets. We sample 9 frames per video which yields a total of 87 633 training images for MOVi-C and 87 741 images on MOVi-E. For evaluation, we use 4 200 frames for MOVi-C and 4 176 frames for MOVi-E from the validation sets in each case. We use a resolution of $128 \times 128$ for both input images and target masks.

**SCANNET and YCB (Yang & Yang, 2022)**  These datasets consist of real-world objects on black backgrounds and were originally introduced to test limitations of object-centric learning methods (Yang & Yang, 2022). SCANNET (originally from Dai et al. (2017)) consists of objects that can be typically be found in indoor scenes (e.g. furniture) and YCB (originally from Calli et al. (2015)) consists of 21 different classes of everyday objects (e.g. food items, kitchen items, tools, etc.). Each of these dataset consist of 10 000 training images and 2 000 evaluation images. Both datasets consist of 2–6 objects per scene. We use a resolution of $128 \times 128$ for both input images and target masks.

**CLEVRTEX (Karazija et al., 2021)**  This is a synthetically constructed dataset where each scene consists of 3–10 simple geometric 3D shapes arranged in a background sampled from a catalogue of 60 different materials. The materials of the objects are also sampled from the same catalogue. This dataset contains 40 000 images for training and 10 000 for validation and test each. We use the 5 000 images from the validation set for our evaluation. CLEVRTEX also offers various OOD splits which utilize materials not seen during training. We do not use these splits; for our zero-shot generalization evaluation, we can directly use the main split since it usually is not a part of the training set we use to train the object-centric model. We use a resolution of $240 \times 240$ for both input images and target masks.

## F   METRICS

### F.1   OBJECT DISCOVERY

**FG-ARI**  The *adjusted rand index* (ARI) measures the similarity between two clusterings (Hubert & Arabie, 1985). We use the instance/object masks as the targets. We only compute this metric for pixels in the foreground (hence, FG-ARI). Unlabeled pixels are treated as background.

**mBO**    To compute the mBO (Pont-Tuset et al., 2017), each predicted mask is assigned to the ground truth mask with highest overlap in terms of IoU. The mBO is computed as the average IoU of these mask pairs.

**Panoptic ARI**    Panoptic ARI is computed as ARI, but uses panoptic mask annotations as ground truth targets. Panoptic masks (Kirillov et al., 2019) provide more detailed mask annotations for an image by assigning a different mask for separate instances of the same object ("things") and also segmenting background regions ("stuff"). We only compute the Panoptic ARI for those images which have at least two masks.

**Panoptic Quality**    The Panoptic Quality (PQ) (Kirillov et al., 2019) is computed by first assigning each predicted mask to the ground truth mask with the highest overlap in terms of IoU, removing all matches that do not have an IoU overlap of at least $0.5$; this results in a unique matching (Kirillov et al., 2019). These mask pairs form the set of true positives (TP). Ground truth masks that were not assigned a predicted mask form the set of false negatives (FN). Similarly, predicted masks that were not assigned to a ground truth mask form the set of false positives (FP). Predicted masks that have an IoU overlap of more than $0.5$ with pixels labeled as "void" or "crowd" are removed from the set of false positives. The panoptic quality is then computed as:

$$PQ = \frac{\sum_{(p,g) \in \text{TP}} \text{IoU}(p,g)}{|\text{TP}| + 0.5|\text{FP}| + 0.5|\text{FN}|} \tag{7}$$

### F.2    DOWNSTREAM PROPERTY PREDICTION

For downstream property prediction, we closely follow the protocol of Dittadi et al. (2022) and Seitzer et al. (2023). In particular, we train linear probes on top of the slots, training them using categorical cross entropy for discrete properties and MSE for continuous properties. Predicted properties for the set of slots are matched to the target set of properties using the Hungarian method, minimizing the total loss of the matching. For discrete properties, we report classification accuracy; for continuous properties, we report the $R^2$-score. We report metrics on the the same test sets used for object discovery.

We designate a portion of the training set as a validation set to measure overfitting; for this purpose, we sample a number of images equal to the size of the test set from the training set. We train for $15\,000$ steps, and evaluate the model with the lowest validation loss. Images without any objects (due to cropping) are filtered out. Following Seitzer et al. (2023), for each image, we only keep the $N$ largest objects, where $N$ equals the number of slots used on that dataset. This way, each object can in principle be matched to a slot.

As properties, we use:

- Location: x- and y-coordinate of the object, using the center-of-mass of the object mask. We report the average $R^2$-score for x- and y-dimension.
- Category: we use the class label with which the objects are annotated on most datasets; on CLEVRTEX, we use the object shape as the category. The used versions of the SCANNET and YCB datasets do not contain any class annotation; as such, we do not report category prediction on these datasets. Note that the number of classes differs per dataset and thus classification results between datasets are not directly comparable. In particular, CLEVRTEX has 4 categories; MOVI-C and MOVI-E have 17 categories; ENTITYSEG has 256 categories.

We report the property prediction for each individual dataset in Table A.7.

## G    EXAMPLES

In this section, we show example predictions for DINOSAUR, SPOT, Slot Diffusion, FT-DINOSAUR, and SAM, where all methods besides SAM were trained on the COCO dataset. FT-DINOSAUR uses a ViT-B/14 encoder with top-k and hi-res adaptation, i.e. the model evaluated in Table 2 and Fig. 7.

- Fig. G.6: in-distribution predictions on COCO.

- Fig. G.7: zero-shot predictions on ENTITYSEG.
- Fig. G.8: transfer predictions on PASCAL VOC.
- Fig. G.9: zero-shot predictions on CLEVRTEX.
- Fig. G.10: zero-shot predictions on MOVi-C.
- Fig. G.11: zero-shot predictions on MOVi-E.
- Fig. G.12: zero-shot predictions on SCANNET.

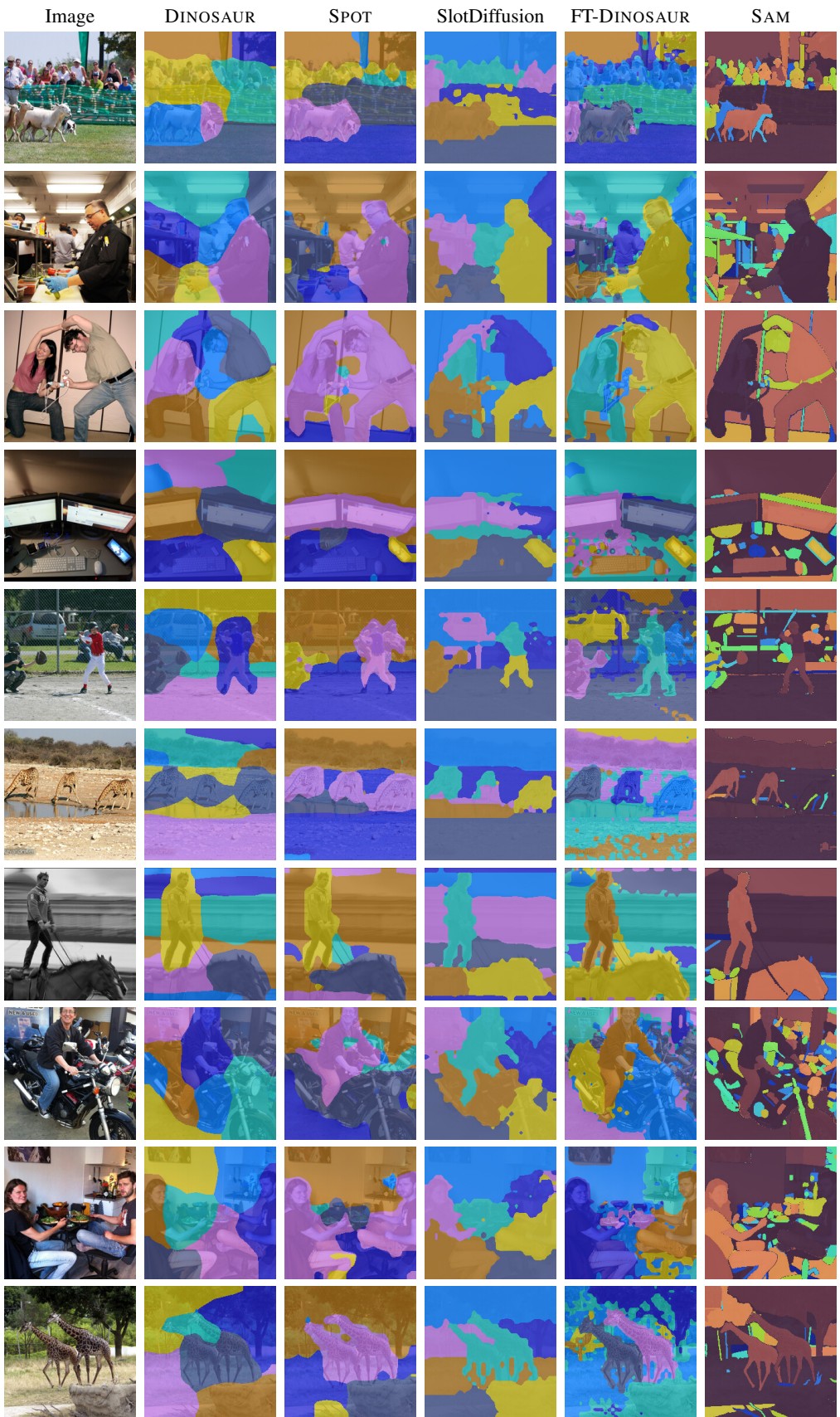

**Figure G.6**: In-distribution examples on Coco.

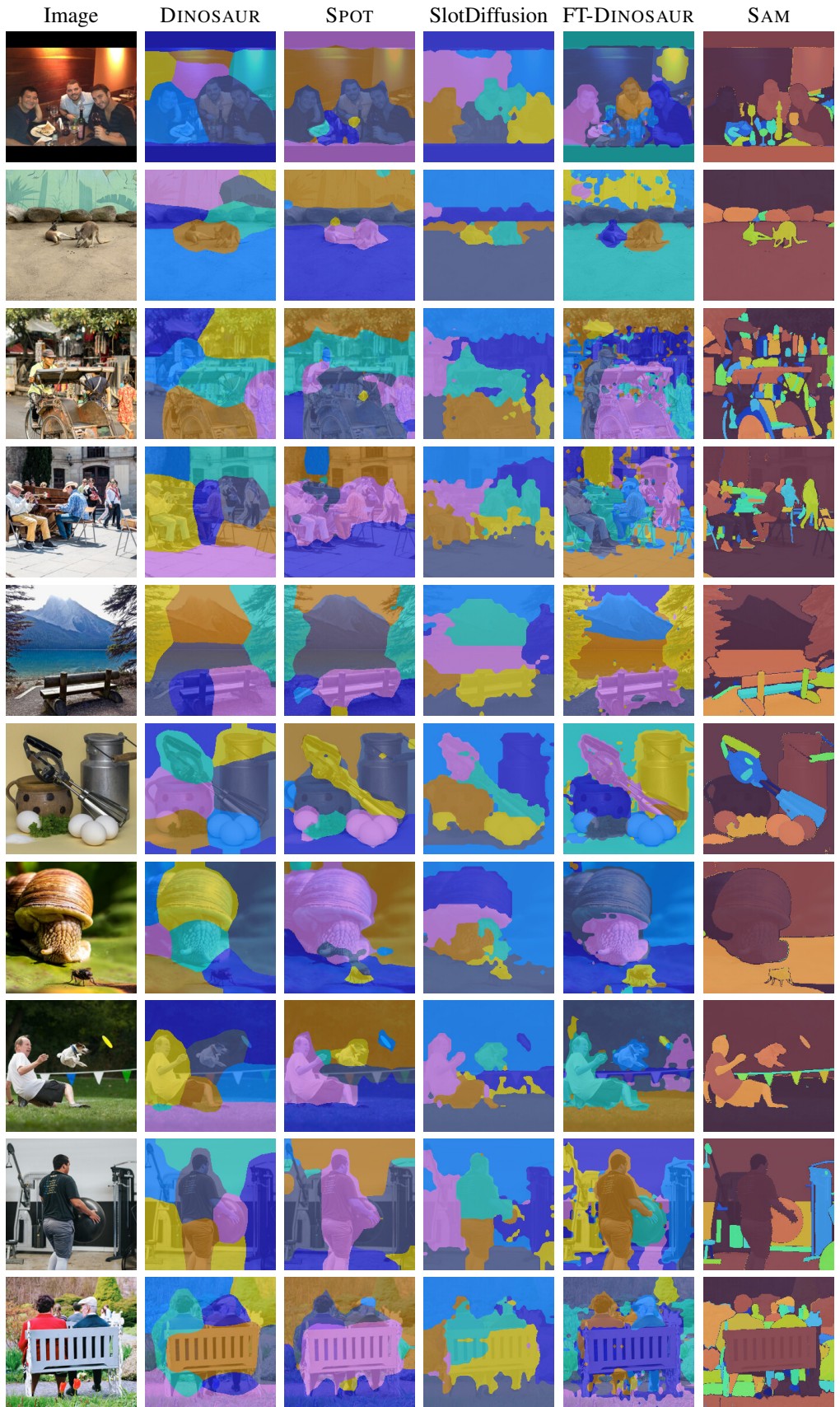

**Figure G.7**: Zero-shot examples on ENTITYSEG.

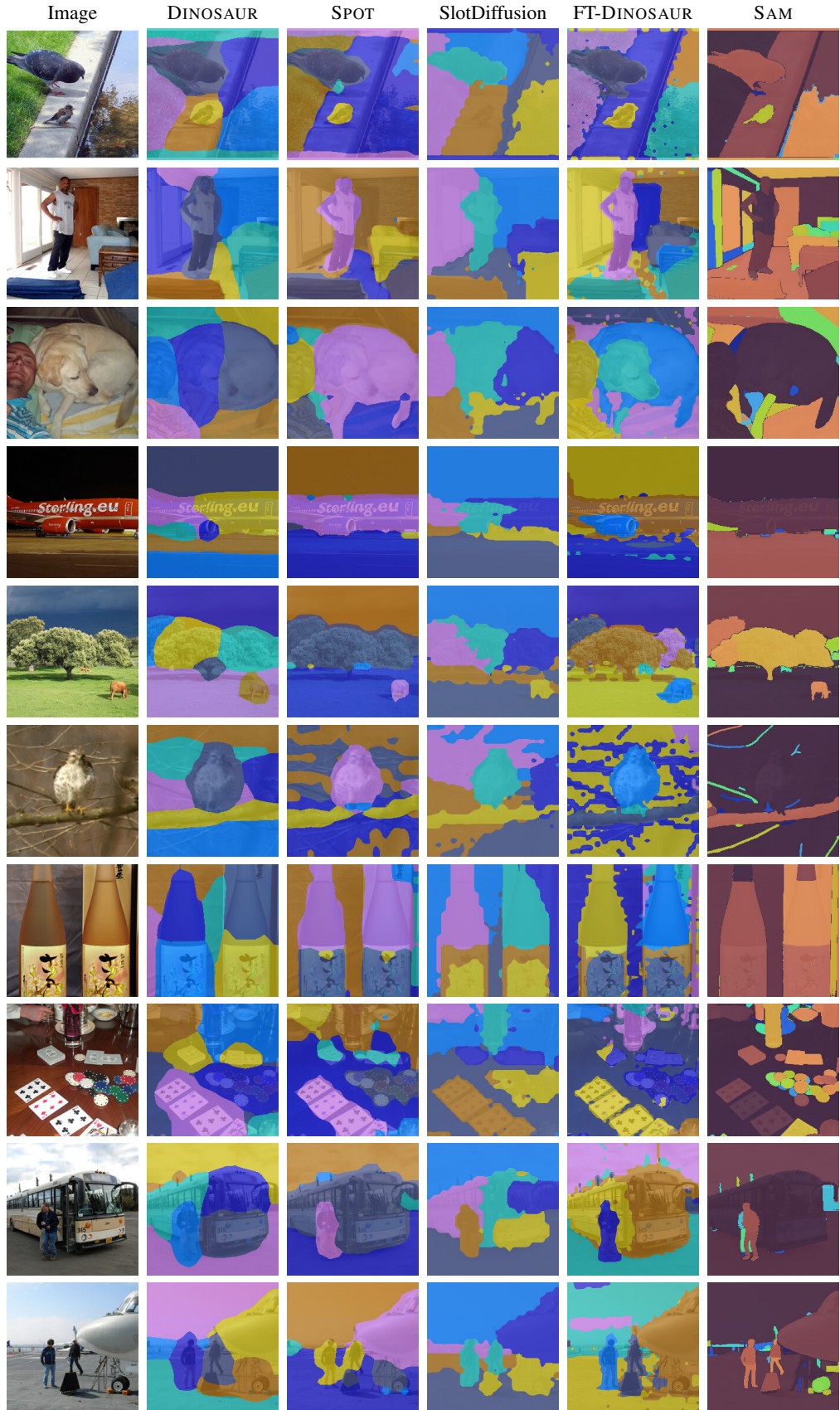

**Figure G.8**: Transfer examples on PASCAL VOC.

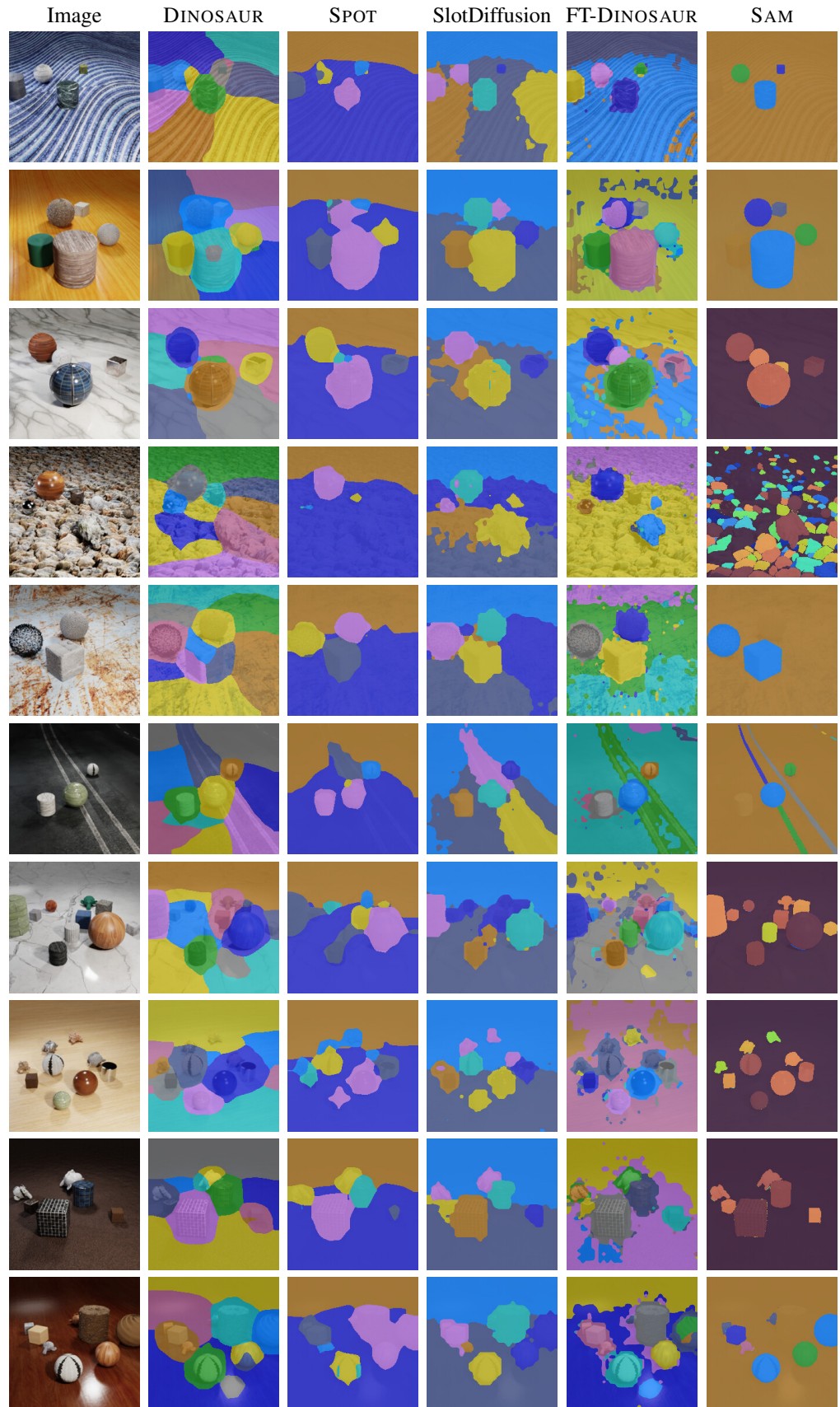

**Figure G.9**: Zero-shot examples on CLEVRTEX.

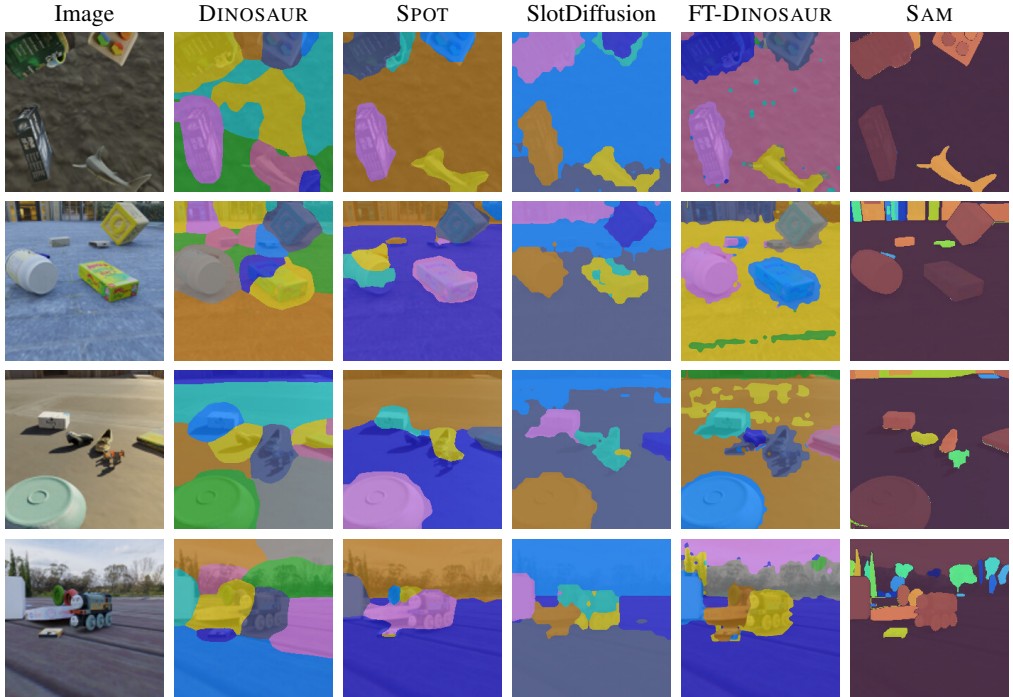

Figure G.10: Zero-shot examples on Movi-C.

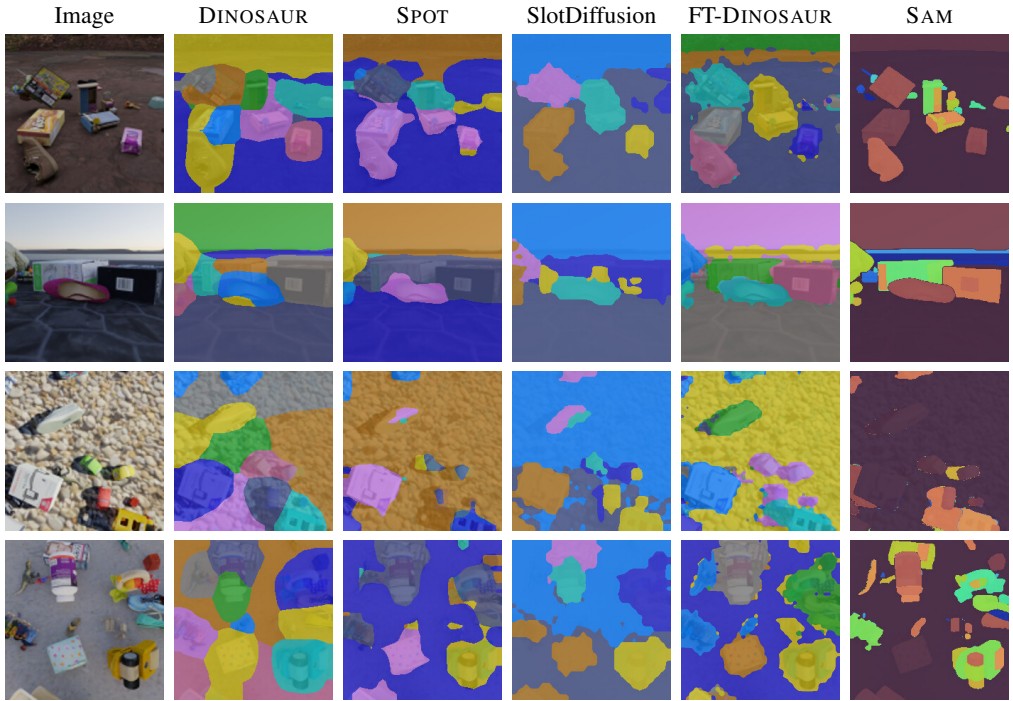

Figure G.11: Zero-shot examples on Movi-E.

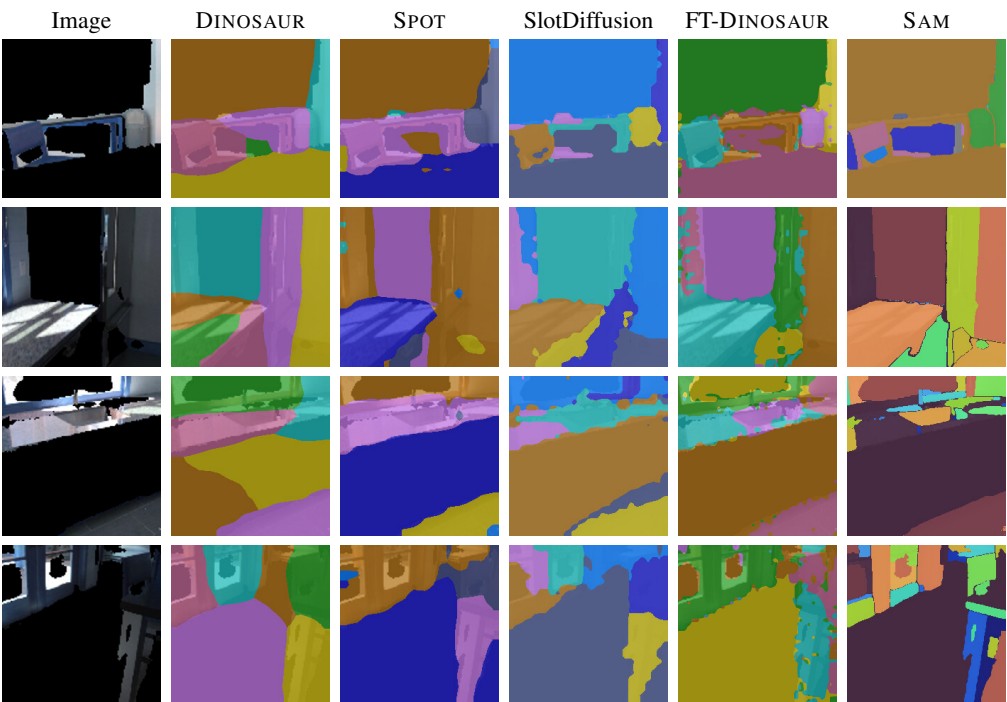

**Figure G.12**: Zero-shot examples on SCANNET.

