# OpenReview forum: "On the Transfer of Object-Centric Representation Learning"
_ICLR.cc/2025/Conference — ICLR 2025 Poster_

### Official Review · Reviewer_neGY · 2024-10-22

**Soundness:** 2
**Presentation:** 3
**Contribution:** 2
**Rating:** 6
**Confidence:** 4

**Summary:**

This paper introduces a benchmark to evaluate the zero-shot generalizability of object-centric learning methods. The benchmark comprises seven synthetic and real-world multi-object datasets, allowing comprehensive testing across different scenarios. Additionally, the paper demonstrates that training on these diverse real-world images significantly enhances the transferability of existing methods to unseen contexts. Moreover, the authors propose an object-centric fine-tuning strategy, building on DINOSAUR’s pre-trained vision encoder, which achieves state-of-the-art performance in unsupervised object discovery with robust zero-shot transfer abilities.

**Strengths:**

- The paper addresses an important topic of exploring zero-shot object-centric learning OOD settings.
- The fine-tuning approach of DINOSAUR improves object-discovery performance and closes the gap between zero-shot and supervised learning.
- Extensive experiments are conducted across various datasets, providing insights into the zero-shot generalizability and scalability of object-centric models.

**Weaknesses:**

My major concern on this paper is regarding the evaluation design:
- The evaluation relies on pre-trained DINO features, which may already possess OOD capabilities due to large-scale pre-training. Since object-centric learning methods using pre-trained encoders only need to focus on feature grouping, these methods can easily transfer learned grouping techniques to new datasets supported by the powerful pre-trained features. As a result, it is difficult to ascertain whether the observed zero-shot generalization stems from the object-centric method itself or DINO’s inherent robustness.
- The paper uses object segmentation as the primary evaluation metric, which might not fully capture the quality of the learned representations. The inclusion of additional downstream tasks, such as object classification or property prediction, would provide a more comprehensive evaluation of the generalizability.

Furthermore, the method proposed in this paper offers limited novelty: the fine-tuning process relies on well-known techniques and does not introduce significant new insights beyond improved hyperparameter tuning.

Lastly, the claim that “training on diverse real-world images improves transferability to unseen scenarios” adds little to the current knowledge, as it is already widely accepted that increased data quantity and diversity generally lead to better model performance.

**Questions:**

See the weaknesses section. Besides, I strongly encourage the authors to conduct experiments without the DINO encoder. Since DINO is pre-trained on ImageNet, which is significantly larger than those used in this paper’s experiments, it's hard to accurately evaluate the model's transferability when using DINO.

---

> ### Author Response · Authors · 2024-11-21
> **Rebuttal Response 1/2**
>
> We thank the reviewer for their feedback! The reviewer generally seems to appreciate our work; their major concerns are 1) the use of pre-trained DINO features while studying the OOD setting, and 2) the lack of downstream tasks besides segmentation. We think 1) may stem from a misunderstanding of the goals of our work, while 2) seems to be an oversight of the reviewer, as we do have experiments with other tasks in the paper. We discuss in more detail in the following.
>
> **About pre-trained DINO features and OOD capabilities**
>
> The reviewer is concerned that the observed robustness stems from the use of pre-trained features. We do agree that the pre-trained encoder likely plays a role for robustness. However, the question of whether the robustness comes from pre-trained features or object-centric components is beside the point: our goal in this paper is **not** to disentangle the source of robustness. Instead, our goal is to analyze the degree of robustness of modern object-centric methods using pre-trained features. From the introduction:
>
> > “While it has been shown that the representations produced by such self-supervised models are fairly robust to changes to the training distribution, it is unclear whether the same holds for an object-centric model trained on top of such representations. Thus, in this work, we study the question of how well object-centric representations using pre-trained features transfer to new data.” (L52)
>
> This is necessary as even if we believe pre-trained DINO features to be robust, (1) they are not object-centric, (2) it is necessary to demonstrate that the combination of object-centric modeling and pre-trained features is still robust. We are the first to study OOD robustness in this context.
>
> Still, we agree that the question where the robustness comes from is interesting. The reviewer also suggests:
>
> > I strongly encourage the authors to conduct experiments without the DINO encoder
>
> Thus, we conducted an additional experiment showing that OCL does exhibit robustness even without a pre-trained encoder. In particular, we expand on the hybrid approach already mentioned in L341: training with a ViT encoder initialized from scratch while predicting DINOv2 features as targets. Note that fully removing the pre-trained targets is not possible as the model would fail to discover objects on real-world data (see the DINOSAUR paper). In this setting, the model is only trained on inputs from the COCO dataset, and we can thus study how well the model transfers OOD to other datasets. We find that this mode, termed as Scratch-DINOSAUR, still achieves a performance of 42.3 FG-ARI, 27.3 mBO on COCO, and the following zero-shot performance (showing FG-ARI, mBO):
>
> Method|MOVi-C|MOVi-E|ScanNet|CLEVRTex|YCB|EntitySeg
> ---|---|---|---|---|---|---
> DINOSAUR|67.0, 34.5|71.1, 24.2|57.4, 40.8|82.5, 35.2|60.2, 42.2|43.5, 19.4
> SlotDiffusion|66.9, 43.6|67.6, 26.4|52.0, 51.7|77.0, 45.0|62.5, 59.2|43.7, 25.1
> SPOT|63.0, 40.8|47.8, 21.5|48.6, 43.2|63.3, 40.0|52.9, 45.1|41.7, 27.4
> FT-DINOSAUR (ViT-S/14)|71.3, 44.2|71.1, 29.9|54.8, 48.4|86.0, 50.1|67.4, 54.5|48.1, 28.1
> Scratch-DINOSAUR (ViT-S/14)|68.3, 31.9|67.9, 23.1|50.5, 36.9|78.8, 31.4|57.3, 35.8|46.6, 22.7
>
> As can be seen, the from scratch-setting still achieves decent transfer performance, mostly matching DINOSAUR (and beating SlotDiffusion and SPOT) in terms of FG-ARI, while falling behind in terms of mBO. This demonstrates that it is **not** necessary to combine a pre-trained encoder with slot attention for zero-shot transfer of object-centric representations. However, combining pre-trained features with our proposed finetuning recipe (FT-DINOSAUR) achieves the best results.
>
> **On additional downstream tasks such as property prediction**
>
> The reviewer appears to have missed that **we do have results on property prediction in the paper**, for example featuring prominently in Figure 1(b) on page 3. Comprehensive results for various models on all datasets are also in the Appendix (Figure A.1, Table A.5). To summarize the results, we find that in general object-centric models can perform well on property prediction on unseen datasets.

---

> ### Author Response · Authors · 2024-11-21
> **Rebuttal Response 2/2**
>
> **Regarding limited novelty of the fine-tuning and lack of insights**
>
> We would like to point out that the finetuning approach combines existing and new techniques, opens new avenues for object-centric models resulting in state-of-the-art performance. While these improvements might seem straightforward, they significantly enhance real-world object-centric learning, revealing that current models can still improve substantially with "engineering" optimizations. These "tricks" are also not so obvious for OCL:
>
> Regarding finetuning: previous works have tried finetuning the pretrained encoder in object-centric models but have either failed (DINOSAUR [1]) or had to use sophisticated tricks which may hurt the model in other ways (SPOT [2]). From the DINOSAUR paper:
>
> > “We [...] test a ViT-B/16 encoder under [...] finetuning DINO pre-trained weights. We find that [...] the finetuning setting fail to yield meaningful objects, resulting in striped mask patterns“ (Sec. 4.3, p. 8)
>
> > "Another option would be further finetuning the pre-trained ViT, but we found that this leads to slots that do not focus on objects. Combining ViT training with Slot Attention might require very careful training recipes.” (footnote of p. 4)
>
> SPOT, only able to finetune the last four layers of the encoder, found that they need a two-stage training procedure with self-regularization and learnable queries from BO-QSA [3] to enable finetuning without any collapse (Table 8 in [2]). We note that learnable queries come with a disadvantage: not being able to vary the number of slots at test time. This hurts especially in the transfer setting (see Table A.4, where SPOT achieves the worst transfer results). In contrast, FT-DINOSAUR provides a simplified finetuning recipe for the full encoder, allowing future research to incorporate finetuning without resorting to complex setups, while additionally benefiting from random queries.
>
> Both *high-resolution adaptation* and *top-k decoding* are novelties in OCL; together, they synergize especially well, as top-k decoding enables training of high-resolution models due to the much reduced GPU memory usage. Top-k decoding also opens up a new design space for decoding (which value for k? Should k be sampled? Different k's between train and test?), which we have not explored in this paper, but is interesting for future work.
>
> Neither of these tricks were known before in OCL. Taken together, these changes are essential for achieving strong performance (Table 1). The key insight this brings is how much improvement can still be gained from careful training and modeling choices, without reaching for more complex changes. Furthermore, by demonstrating that finetuning the encoder is possible, we hope to encourage future work to adopt this design; previous models based on DINOSAUR were mostly using a frozen (and suboptimal, as we show) encoder by inertia.
>
> [1] Bridging the Gap to Real-World Object-Centric Learning
>
> [2] SPOT: Self-Training with Patch-Order Permutation for Object-Centric Learning with Autoregressive Transformers
>
> [3] Improving Object-centric Learning with Query Optimization
>
> **Regarding limited new knowledge from our study**
>
> The reviewer suggests that our claim "training on diverse real-world data improves transferability to unseen scenarios" offers no new knowledge. First, we admit that we phrased this slightly carelessly in the abstract by including the word "diverse", but this is actually not the main conclusion we drew from our study. Instead, our findings indicate that "diversity" is not necessarily the primary factor in zero-shot performance: "zero-shot performance is fairly similar when trained on COCO and EntitySeg (high diversity, many objects) or PASCAL VOC (less diversity, few objects)" (L256). Our conclusion was instead that "training on complex natural data is an important component for zero-shot transfer" (L295). We will change the abstract accordingly.
>
> While this may seem obvious in deep learning, there was no empirical evidence for it in the object-centric literature, which typically focused on the in-distribution setting. We are the first to explore in-the-wild zero-shot object-centric learning, demonstrating that models generalize to unseen scenes and domains when trained on real-world data.
>
> Our experiments reveal that training on natural data (COCO, EntitySeg, VOC) enables object-centric models to discover unseen objects in unseen backgrounds (ClevrTex, MOVi-C, MOVi-E). This generalization ability was previously unknown. Additionally, our results show that the models generalize robustly when varying the number of slots during inference, irregardless of being trained with a fixed (and different) number of slots during training.
>
> Finally, we stress the importance of empirical evidence, even for seemingly trivial claims. For instance, while "scaling data size improves performance" might appear obvious, our experiments (Sec 3.3, Fig. 2b) demonstrate that this is not always the case.

---

> ### Comment · Reviewer_neGY · 2024-11-23
> **Response to Author Rebuttal**
>
> Thank you to the authors for their detailed rebuttal. I appreciate the effort put into addressing my concerns, and I believe most of my concerns have been addressed. While I still maintain some reservations regarding the depth of insight presented in the paper, I acknowledge the empirical success of the proposed method, which demonstrates its practical value.
>
> Based on these considerations, I am now slightly inclined to support accepting this paper and have raised my rating to 6.

---

### Official Review · Reviewer_dN2g · 2024-10-30

**Soundness:** 3
**Presentation:** 3
**Contribution:** 3
**Rating:** 5
**Confidence:** 3

**Summary:**

This paper studies the zero-shot performance of object-centric representation learning. The authors first combine 7 existing datasets to establish a comprehensive testbed, enabling them to evaluate the transferability of three recently proposed object-centric learning models. Based on the experimental findings, they propose finetuning the pre-trained encoders on the COCO dataset to enhance zero-shot performance on their testbed.

**Strengths:**

- The paper is well-motivated, beginning with a clear question and proposing a testbed to evaluate the performance of current methods. Based on the evaluation results, the authors propose method to further enhance the model.

- The proposed finetuning method is straightforward yet effectively addresses model collapse problem

**Weaknesses:**

1. I would consider this as an analysis paper. However, many of the conclusions appear somewhat obvious. For instance, SAM already demonstrates strong zero-shot transfer capabilities in object-centric models, even extending well to domains like medical imaging. Additionally, it’s widely understood that training on complex natural datasets is crucial for zero-shot transfer performance, and using real data is key to enhancing this ability.
This paper would be more interesting with additional and more detailed experiments and insights provided by the testbed. For example, is it possible to identify some common failure cases and explore possible reasons behind these failures? I guess that could offer more insights.
2. The paper proposes to finetune on an additional dataset (COCO) to enhance zero-shot performance. However, it doesn't have any specific design for zero-shot task. Additionally, task-specific finetuning on a general pretrained model typically leads to performance improvements.

**Questions:**

Please see the weaknesses section.

A small question: Would it be possible to provide a more detailed comparison of the similarities between COCO and the 7 datasets in testbed?

---

> ### Author Response · Authors · 2024-11-21
> **Rebuttal Response (1/2)**
>
> We thank the reviewer for their comments. We address their concerns below.
>
> **SAM already demonstrates strong zero-shot transfer capabilities in object-centric models.**
>
> While SAM does exhibit zero-shot transfer, it is a segmentation model and not what would be called an object-centric (representation learning) model. An object-centric model is primarily learning distinct representations for different objects in a scene. While segmentation can be a use case for an object-centric model, it is not its main use case. More interesting are applications that benefit from structured representations. For example, we use the representations for property prediction in the paper (Fig. 1b and Table A.5). Furthermore, there are many downstream applications of object-centric representations such as reinforcement learning [1, 2], world modeling [3], visual question answering [3, 4], and image generation [4]. SAM cannot be used in any of these cases since it does not output representations, it only outputs masks. While we do compare with SAM for object discovery, we mainly use it as an upper bound in terms of segmentation performance for our approach.
>
> Another important distinction between SAM and object-centric learning is that SAM is trained supervised while object-centric learning is fully unsupervised. This makes it more difficult to finetune SAM on new domains and in combination with new tasks.
>
> Our goal in the paper is to demonstrate that object-centric models also exhibit zero-shot transfer. This is necessary because 1) SAM does not cover all (or even most) important use cases for object-centric models, and 2) SAM is an entirely different class of model, from whose capabilities we can not infer the capabilities of object-centric models.
>
> - [1] An Investigation into Pre-Training Object-Centric Representations for Reinforcement Learning https://arxiv.org/abs/2302.04419
> - [2] Self-supervised Visual Reinforcement Learning with Object-centric Representations https://arxiv.org/abs/2011.14381
> - [3] SlotFormer: Unsupervised Visual Dynamics Simulation with Object-Centric Models https://arxiv.org/abs/2210.05861
> - [4] SlotDiffusion: Object-Centric Generative Modeling with Diffusion Models https://arxiv.org/abs/2305.11281
>
> **It’s widely understood that training on complex natural datasets is crucial for zero-shot transfer performance, and using real data is key to enhancing this ability**
>
> While this statement may seem trivial in the current theme of deep learning research, there was no empirical evidence for it in the object-centric literature, which typically focused on the in-distribution setting. We are the first to explore in-the-wild zero-shot object-centric learning, demonstrating that models generalize to unseen scenes and domains when trained on real-world data.
> Our experiments reveal that training on natural data (COCO, EntitySeg, VOC) enables object-centric models to discover unseen objects in unseen backgrounds (ClevrTex, MOVi-C, MOVi-E). This generalization ability was previously unknown. Additionally, our results show that the models generalize robustly when varying the number of slots during inference, irregardless of being trained with a fixed (and different) number of slots during training.
>
> Finally, we stress the importance of empirical evidence, even for seemingly trivial claims. For instance, while "*scaling data size improves performance*" might appear obvious, our experiments (Sec 3.3, Fig. 2b) demonstrate that this is not always the case.
> Is it possible to identify some common failure cases and explore possible reasons behind these failures
> We do present several categories of common failure cases in Fig. B.5, and discuss them in App. B (with possible resolutions). Furthermore, the negative data scaling result can be considered as a failure case of existing object-centric models (Fig 2b). If the reviewer has any other suggestions for specific analysis we could do, we would be happy to try to address them.
>
> **the paper doesn't have any specific design for zero-shot task**
>
> We interpret the zero-shot task in the context of object-centric learning as follows:
> > “For our benchmark, we interpret zero-shot transfer to mean that a model can successfully discover objects of novel categories not occurring in the training data” (L155)

---

> ### Author Response · Authors · 2024-11-21
> **Rebuttal Response (2/2)**
>
> **task-specific finetuning on a general pretrained model typically leads to performance improvements**
>
> We generally agree with the reviewer. However we would like to point out that 1) it is not obvious that DINOv2 features can be further improved for object discovery, as they are generally considered to already exhibit a strong object focus, 2) it is not at all obvious that features finetuned on COCO do still exhibit zero-shot transfer, and 3) finetuning in the context of object-centric models is tricky, as evident by these quotes from the DINOSAUR paper [1]:
>
> > We [...] test a ViT-B/16 encoder under [...] finetuning DINO pre-trained weights. We find that [...] the finetuning setting fail to yield meaningful objects, resulting in striped mask patterns“ (section 4.3, p. 8)
>
> > Another option would be further finetuning the pre-trained ViT, but we found that this leads to slots that do not focus on objects. Combining ViT training with Slot Attention might require very careful training recipes.” (footnote of p. 4)
>
> We address 3) by providing a stable recipe for finetuning that considerably pushes the state-of-the-art in real-world object-centric learning, yielding the essential insight that there is ample room left for improvement in current OCL models “just” by using engineering tricks. We do address 1) and 2) by extensively experimenting and analyzing the resulting model (Sec. 4.2, Table 1, Sec. 5).
>
> [1] Seitzer et al. Bridging the Gap to Real-World Object-Centric Learning https://openreview.net/forum?id=b9tUk-f_aG
>
> **Would it be possible to provide a more detailed comparison of the similarities between COCO and the 7 datasets in testbed?**
>
> We thank the reviewers for this question! We have already provided some discussion in App. E, and we elaborate further in this rebuttal.
>
> - COCO consists of 118k images which contain objects spanning 91 different categories. These images are mostly real-world images representing natural scenes. We typically run slot attention with 7 slots for COCO, this 7 represents the ideal average number of objects in COCO scenes.
>
> - EntitySeg: this dataset also contains natural scenes but it is significantly more complex than COCO as it contains 256 total object categories many of them not present in COCO. Therefore, for a model trained on COCO, these categories are out-of-distribution. However, our experiments demonstrate that models trained on COCO still work well on EntitySeg illustrating the transferability of our models.
>
> - ClevrTex: this is a synthetic dataset which contains objects of 3 different sizes, 4 different shapes and 60 different materials. All these are synthetically generated and very different from the natural real-world scenes of COCO.
>
> - MOVi-C and MOVi-E: these datasets are generated by synthetically placing real-world objects in real-world backgrounds. They are created such that the scenes are cluttered with objects. These datasets contain 17 categories of objects. MOVi-C contains up to 10 objects per scene, while MOVi-E contains up to 20 objects per scene.
>
> - Scannet: this dataset contains images of indoor environments - specifically houses. While coco mainly contains outdoor scenes, scannet presents indoor scenes.
>
> - YCB: this dataset contains images of objects typically found in the house such as scissors, cans etc.
>
> Therefore, EntitySeg, ClevrTex, Scannet, and YCB are all significantly different from COCO and test whether object-centric models trained on COCO can adapt to new visual domains. On the other hand, MOVi-C and MOVi-E evaluate whether models trained on COCO can adapt to scenes with many more objects than those seen during training.

---

> > ### Author Response · Authors · 2024-11-25
> > **Follow up**
> >
> > We thank the reviewer for their insightful review. As the discussion period draws to a close we would like to follow up to check whether we addressed all the concerns which the reviewer had and whether they would consider updating their rating of the paper accordingly. Furthermore, we would be happy to address any further concerns raised by the reviewer.
> >
> > We would also like to mention about the additional experiments we have performed to address concerns raised by other reviewers.
> >
> > - https://openreview.net/forum?id=bSq0XGS3kW&noteId=5g0aO6XPat - Reviewer neGY asked for experiments wherein we train the model from scratch instead of finetuning a pretrained backbone. We found that the proposed techniques introduced in this paper can be used to train object-centric models from scratch which exhibit strong zero-shot generalizability. Reviewer neGY has correspondingly raised score from 3 to 6.
> > - https://openreview.net/forum?id=bSq0XGS3kW&noteId=v71FMhyhp8 - Reviewer SC2i has asked about using a learnable init to prevent instabilities during training. We performed this experiment and found that FT-Dinosaur trained with learnable init. exhibits weak zero-shot transfer similar to SPOT which also has learnable initialization. This can be attributed to the inability to use variable number of slots during inference when using a learnable initialization.

---

> > > ### Comment · Reviewer_dN2g · 2024-11-27
> > >
> > > Thank you for the detailed reply. I have carefully reviewed your responses, as well as the feedback and discussions from other reviewers. I appreciate the effort and the empirical results presented. However, I still feel that the proposed method lacks novelty, and the conclusions seem somewhat evident and supported by previous work, such as the zero-shot transfer capabilities of SAM or DINO for various tasks. Therefore, I would like to maintain my current rating.

---

### Official Review · Reviewer_ixZS · 2024-11-02

**Soundness:** 2
**Presentation:** 1
**Contribution:** 1
**Rating:** 3
**Confidence:** 4

**Summary:**

This paper discusses the transferability of self-supervised models, such as DINO, and the factors about the transferability performance. Then, the authors further explore the finetuning strategy to adapt the model to conduct object discovery task.

**Strengths:**

+ The topic about object discovery task is interesting.

**Weaknesses:**

The writing is confusing and difficult to understand. Not only the expression of writing is confusing, but also many important clarifications about technique can not be found in the paper. We list these issues as follows.
+ The evaluation of zero-shot transfer. In the main body of paper, the authors only list the datasets and metrics used for evaluation, but don't mention why and how.
+ For object centric finetuning, the authors don't mention how to conduct slot attention and top-k mlp decoder (the structure details) and why use these module. What is the loss function? Where do the output images come from in Figure 3? Are the output masked images just the output of DINOv2?
In summary, I am inclined to reject this submission in the current presentation.

**Questions:**

See Weaknesses.

---

> ### Author Response · Authors · 2024-11-21
> **Rebuttal Response**
>
> We thank the reviewer for their review. However, we respectfully disagree that the presentation of the paper is lacking. As evidence, we would like to point the reviewer to other reviews of our paper, rating our presentation as excellent (Score 4, Rev. SC2i) or good (Score 3, Revs. dN2g and neGY). We have clarified some of the questions asked by the reviewer below.
>
> **The evaluation of zero-shot transfer. In the main body of paper, the authors only list the datasets and metrics used for evaluation, but don't mention why and how.**
>
> These discussions are already present in the paper. The datasets are described in more detail in App. Sec. E (referenced in Sec. 3.1 (Evaluation Datasets, Page 4)). The metrics are described in App. Sec. F (referenced in Sec. 3.1 (Metrics, Page 4)). Note that these are standard datasets and metrics used in object-centric learning, hence we shifted their descriptions to the appendix while focusing on our main contributions (zero-shot benchmark and finetuning) in the main paper. We would be happy to answer any further questions which the reviewer may have about the datasets or the metrics.
>
> **For object centric finetuning, the authors don't mention how to conduct slot attention and top-k mlp decoder (the structure details) and why use these module. What is the loss function? Where do the output images come from in Figure 3? Are the output masked images just the output of DINOv2? In summary, I am inclined to reject this submission in the current presentation.**
>
> - Slot attention is currently the main module used for extracting object-centric representations in the object-centric literature. As it is widespread and standard, we did not see the need to include a description of slot attention in the paper. To address the reviewer’s concern, we have now included a description in App. Sec. C.2 in the revision of the paper.
>
> - Top-k decoding is described in detail in App. Sec. C.3, with the motivation clearly explained in Sec. 4.1, L355-363.
> FT-DINOSAUR is directly based on the DINOSAUR model [1]; for sake of brevity, we did not expand on DINOSAUR again in the paper. We apologize for this omission. We have now added a section describing the DINOSAUR approach in App. Sec. C.1.
>
> - The output masks are obtained from the Top-K MLP decoder which outputs masks and predicted values for each slot. Note that it is standard for the decoders of object-centric models to output object masks. We have now added a description of this process in App. Sec. C.1 (last paragraph).
>
> - The output masked images are not the output of DINOv2, as DINOv2 does not output object masks. Instead, an object-centric module (slot attention) processes DINOv2 features to obtain a set of slots as shown in Figure 3. These slots are decoded by the Top-K MLP decoder to produce a set of masks per patch and the corresponding patch values.
>
> If the reviewer has any other questions, or further concerns regarding our technical and research contributions, we would be happy to address them.

---

> > ### Author Response · Authors · 2024-11-25
> > **Follow up**
> >
> > We thank the reviewer for their insightful review. As the discussion period draws to a close we would like to follow up to check whether we addressed all the concerns which the reviewer had and whether they would consider updating their rating of the paper accordingly. Furthermore, we would be happy to address any further concerns raised by the reviewer.
> >
> > We would also like to mention about the additional experiments we have performed to address concerns raised by other reviewers.
> >
> > - https://openreview.net/forum?id=bSq0XGS3kW&noteId=5g0aO6XPat - Reviewer neGY asked for experiments wherein we train the model from scratch instead of finetuning a pretrained backbone. We found that the proposed techniques introduced in this paper can be used to train object-centric models from scratch which exhibit strong zero-shot generalizability.  Reviewer neGY has correspondingly raised score from 3 to 6.
> > - https://openreview.net/forum?id=bSq0XGS3kW&noteId=v71FMhyhp8 - Reviewer SC2i has asked about using a learnable init to prevent instabilities during training. We performed this experiment and found that FT-Dinosaur trained with learnable init. exhibits weak zero-shot transfer similar to SPOT which also has learnable initialization. This can be attributed to the inability to use variable number of slots during inference when using a learnable initialization.

---

### Official Review · Reviewer_SC2i · 2024-11-04

**Soundness:** 4
**Presentation:** 4
**Contribution:** 3
**Rating:** 6
**Confidence:** 4

**Summary:**

This paper studies the zero-shot transfer capabilities of object-centric representation learning models and proposes improvements to enhance their generalization. The key contributions are:
- A benchmark comprising 7 diverse datasets to evaluate zero-shot transfer of object-centric models.
- A novel finetuning approach that adapts pre-trained vision encoders for object discovery.
- The method proposed in this paper achieves state-of-the-art results for object discovery tasks on both in-distribution and out-of-distribution scenarios.

**Strengths:**

- Technical Innovation:
	- First systematic study of zero-shot transfer in object-centric learning
	- Novel finetuning strategy that successfully adapts pre-trained encoders
- Adequate experimentation and analysis：
	- Comprehensive empirical evaluation across multiple datasets and metrics
	- Thorough ablation studies validating each component
- Clear presentation:
	- Well-structured and clearly written
	- Comprehensive appendix with implementation details

**Weaknesses:**

- The proposed method lacks novel and essential insights.  The fine-tuning strategy, high-resolution adaptation, and top-k decoding are engineering improvements that come easily to mind.

- The paper shows that current models don't scale well with data size (Fig 2b), especially for real-world data, but doesn't propose any analyses or solutions for this limitation.

- These datasets, while diverse, are still relatively small-scale for pre-training compared to modern vision datasets. The ScanNet and YCB datasets used in [1] provide images without background. Only COCO, PASCAL, and EntitySeg are more consistent with the natural image distribution. This may be one reason why no evident scaling law has been observed.

[1] Promising or Elusive? Unsupervised Object Segmentation from Realworld Single Images.

**Questions:**

1. I'd like to discuss the "blockwise exponentially decaying learning rates". According to the ablation studies (Tab. 1),  blockwise learning rates don't bring significant improvement. The authors propose it because "the encoder would initially drift away from its pre-trained initialization, likely induced by the noisy gradients from the randomly initialized slot attention module". Since this phenomenon may be caused by the randomness of slot initialization, why not consider changing the initialization method of slots? Why not initialize slots with learnable queries like BO-QSA [1], OSRT [2], and SPOT [3], which have been proven to be effective? Instead, this paper introduces "blockwise exponentially decaying learning rates".  If initializing slots with learnable queries does not solve this problem, the problem may not be caused by random slot initialization.

2. I'm interested in why these models don't scale well with dataset size, especially on real-world datasets such as COCO. What might help to improve scaling behavior? If the authors find out in their experiments what factors help enhance scaling behavior, I believe it is important to highlight them to provide more insight into the progress in this field, which can also increase the impact of this work.

[1] Improving Object-centric Learning With Query Optimization.
[2] OSRT: Object Scene Representation Transformer.
[3] SPOT: Self-Training with Patch-Order Permutation for Object-Centric Learning with Autoregressive Transformers.

---

> ### Author Response · Authors · 2024-11-21
> **Rebuttal Response (1/3)**
>
> We thank the reviewer for their detailed review and appreciating the presentation of our paper. We address the reviewer's comments below.
>
> **The proposed method lacks novel and essential insights. The fine-tuning strategy, high-resolution adaptation, and top-k decoding are engineering improvements that come easily to mind.**
>
> While these improvements may seem obvious, they are a) not so obvious for object-centric learning, and b) together considerably push the state-of-the-art in real-world object-centric learning, yielding the essential insight that there is ample room left for improvement in current OCL models “just” by using engineering tricks.
>
> We comment on the individual “tricks” in more detail:
>
> Regarding *finetuning*: previous works have tried finetuning the pretrained encoder in object-centric models but have either failed (DINOSAUR [1]) or had to use sophisticated tricks which may hurt the model in other ways (SPOT [2]). From the DINOSAUR paper:
>
> > “We [...] test a ViT-B/16 encoder under [...] finetuning DINO pre-trained weights. We find that [...] the finetuning setting fail to yield meaningful objects, resulting in striped mask patterns“ (section 4.3 page 8)
>
> > “Another option would be further finetuning the pre-trained ViT, but we found that this leads to slots that do not focus on objects. Combining ViT training with Slot Attention might require very careful training recipes.” (footnote of page 4)
>
> SPOT, only able to finetune the last four layers of the encoder, found that they need a two-stage training procedure with self-regularization and learnable queries from BO-QSA [3] to enable finetuning without any collapse (Table 8 in [2]). We note that learnable queries come with a disadvantage: not being able to vary the number of slots at test time. This hurts especially in the transfer setting (see Table A.4, where SPOT achieves the worst transfer results). In contrast, FT-DINOSAUR provides a simplified finetuning recipe for the full encoder, allowing future research to incorporate finetuning without resorting to complex setups, while additionally benefiting from random queries.
>
> Both *high-resolution adaptation* and *top-k decoding* are novelties in the object-centric literature; together, they synergize especially well, as top-k decoding enables training of high-resolution models due to the much reduced GPU memory usage. Top-k decoding also opens up a new design space for decoding (which value for k? Should k be sampled? Different values for k between train and test?), which we have not explored in this paper, but is interesting for future work.
>
> Neither of these tricks were known or used before in object centric literature. Taken together, these changes are essential for achieving strong performance (Table 1). The key insight this brings is how much improvement can still be gained from careful training and modeling choices, without reaching for more complex changes. Furthermore, by demonstrating that finetuning the encoder is possible, we hope to encourage future work to adopt this design; previous models based on DINOSAUR were mostly using a frozen (and suboptimal, as we show) encoder by inertia.
>
> - [1] Seitzer et al. Bridging the Gap to Real-World Object-Centric Learning https://openreview.net/forum?id=b9tUk-f_aG
> - [2] SPOT: Self-Training with Patch-Order Permutation for Object-Centric Learning with Autoregressive Transformers https://arxiv.org/abs/2312.00648
> - [3] Improving Object-centric Learning with Query Optimization https://arxiv.org/abs/2210.08990

---

> ### Author Response · Authors · 2024-11-21
> **Rebuttal Response (2/3)**
>
> **The paper shows that current models don't scale well with data size (Fig 2b), especially for real-world data, but doesn't propose any analyses or solutions for this limitation.**
>
> We do not think that this experiment should be counted as a weakness of our work. Our goal in this paper was not to build an object-centric model which scales well in data size, but to analyze the transfer behavior of current real-world object-centric learning models. As part of this analysis, we investigated the influence of the size of the training data and decided that the lack of scaling was an important finding to share with the community.
>
> Of course, it would be great to overcome this lack of scalability, and we hope that our findings can inspire the community to work on it! For this paper, we consider this issue to be out-of-scope.
>
> **I'm interested in why these models don't scale well with dataset size, especially on real-world datasets such as COCO.**
>
> This is a key question! While we do not have a definitive answer, our intuition is that the principles behind current approaches to object-centric learning are at odds with scalability. Object-centric models rely on strong inductive biases to discover objects, which are mainly implemented by tight bottlenecks (in the form of the slots representation, but also the limited expressivity of slot attention and the (MLP) decoder). The use of these bottlenecks essentially hurts the model’s ability to absorb  data, leading to the plateauing behaviors observed in Fig. 2b.
>
> To demonstrate that strong bottlenecks are needed for object-centric learning, we perform an experiment where we incrementally make the bottleneck in DINOSAUR weaker by progressively increasing the slot dimension from 256 to 2048. We find that performance drops as we increase the slot dimension.
>
> | Slot Dim | 256 | 1024 | 2048 | 4096 |
> | ---- | ---- | ---- | --- | ---- |
> |ARI | 42.5 | 41.6 | 35.4 | 16.9 |
>
> **What might help to improve scaling behavior?**
>
> Following the previous intuition, the question is how model capacity can be increased while retaining enough inductive biases for object discovery. One approach we already demonstrated in the paper is to finetune the encoder features (increasing capacity) through the slot bottleneck; in Fig. A.3, we show that this already leads to (somewhat) better scalability than for DINOSAUR.
>
> Going further, another approach could be to switch to softer inductive biases, maybe in the form of suitable regularization losses, that shape the behavior of a high-capacity model (e.g. a standard Transformer) to discover objects. We conjecture that the resulting models may lose some of the segmentation quality of current models, but that their representations might ultimately scale better.
>
> **These datasets, while diverse, are still relatively small-scale for pre-training compared to modern vision datasets.**
>
> We agree with the reviewer that these dataset are not large by modern standards. However, they constitute a representative set of challenging datasets that the object-centric community has considered in the past (plus the even more challenging EntitySeg).
>
> Also, the goal of our paper was not to push pre-training, but to show that existing pre-trained models can be finetuned with very little data to obtain strong transferable object-centric features. It would be very interesting to consider larger datasets for pre-training in future work.

---

> ### Author Response · Authors · 2024-11-21
> **Rebuttal Response (3/3)**
>
> **About blockwise exponentially decaying learning rates and learnable slot initialization**
>
> First, we would like to clarify a potential misunderstanding of the reviewer:
>
> > The authors propose it because "the encoder would initially drift away from its pre-trained initialization, likely induced by the noisy gradients from the randomly initialized slot attention module" Since this phenomenon may be caused by the randomness of slot initialization, why not consider changing the initialization method of slots?
>
>
> The random initialization in the quote from the paper does not refer to the slot initialization, but rather to the weights of the slot attention module at the beginning of training.
>
> That said, random slot initializations may still negatively affect finetuning through noise. To verify this, we perform the ablation mentioned by the reviewer. We use learnable initial slots and do not use blockwise exponential decaying learning rates. This approach achieves 44.3 ARI and 29.9 mBO on COCO as compared to 48.5 ARI and 30.7 mBO achieved by the proposed approach. Further, we report the zero-shot performance of this model on all the datasets in the table below. We denote this model as FT-DINOSAUR (learnable init.), and show FG-ARI, mBO in the following table:
>
>
> | Method                                                                 | Movi-C   | Movi-E   | ScanNet  | CLEVRTex | YCB      | EntitySeg |
> |------------------------------------------------------------------------|----------|----------|----------|----------|----------|-----------|
> | **FT-DINOSAUR (learnable init.)**                                      | 72.2, 30.8 | 65.6, 18.2 | 54.9, 42.9 | 81.7, 31.8 | 67.3, 42.8 | 47.7, 24.2 |
> | FT-DINOSAUR (ViT-S/14) (w/ random init. and blockwise exponential lr decay) | 71.3, 44.2 | 71.1, 29.9 | 54.8, 48.4 | 86.0, 50.1 | 67.4, 54.5 | 48.1, 28.4 |
> | FT-DINOSAUR (ViT-B/14) (w/ random init. and blockwise exponential lr decay) | 73.3, 42.9 | 69.7, 27.9 | 55.8, 48.6 | 83.9, 45.9 | 70.1, 54.1 | 49.7, 29.0 |
>
> We can see that while this model is competitive in terms of ARI, it achieves much worse performance than the proposed FT-DINOSAUR (w/ random init. and blockwise exponential lr decay) in terms of mBO which means that it fails to produce sharp masks. This shows the importance of blockwise exponential lr decay and random initialization. Note also that learnable slot init also comes with the disadvantage of not being able to change the number of slots at test time, which is important in the zero-shot setting.

---

> > ### Comment · Reviewer_SC2i · 2024-11-24
> > **Response to Author Rebuttal**
> >
> > Thank you for this comprehensive rebuttal. The authors have effectively addressed most of my concerns. Given the extensive experiments exploring this field and the method's demonstrated empirical effectiveness, I support acceptance of this paper. Since it lakes deep insights and innovation, I maintain my rating of 6.

---

### Meta-Review · Area_Chair_zCXT · 2024-12-21

**Metareview:**

The reviewers appreciated well-motivated and novel research direction and the associated benchmark, strong performance of the proposed method, extensive experiments and analyses, and clarity of the paper. They at the same time unanimously pointed out the lack of novel and essential insights (i.e., too obvious conclusions) and incremental novelty of the proposed method (i.e., presenting engineering tricks rather than making breakthrough). In addition, they raised concerns with missing analysis or solution for the scalability issue (SC2i), small scales of the datasets used in the paper (SC2i), unclear motivation and benefits of the learning rate scheduling strategy (SC2i), no design element for zero-shot transfer (dN2g), and the design of the evaluation protocol (neGY).

The authors' rebuttal and subsequent responses in the discussion period address many of these concerns but failed to fully assuage all of them. In particular, the reviewers still pointed out the lack of technical novelty and too obvious conclusion after the discussion period. As a result, the reviewers shared nearly identical opinions about the paper's strengths and weaknesses: they appreciated the value of zero-shot object-centric learning and the strong performance of the proposed method, yet at the same time raised common concerns with the lack of novelty in insight and technique. However, their final ratings were mixed depending on which parts of the paper they valued more: two borderline accepts and one borderline reject.

The AC agrees that the conclusion of benchmarking in this paper is a bit trivial and that the proposed method is more like a bag of engineering tricks. However, the AC found that the paper is valuable nevertheless; it provides essential tricks for improving object-centric learning, which have not been known in the field so far and will be a cornerstone for follow-up studies in object-centric learning.
Also, the findings of this paper will open a variety of research questions and directions. Last, but not least, the benchmark is valuable too.

The AC thus recommends acceptance of the paper. The authors are strongly encouraged to carefully revise the paper to reflect the valuable comments by the reviewers and to add new results brought up in the rebuttal and discussions.

**Additional Comments On Reviewer Discussion:**

The AC disregarded the review of ixZS, the second reviewer, when making the final decision since it is very short, vague, and uninformative. To be more specific, the main objection of this reviewer was weak clarity of the manuscript, but the AC found that the paper is well written, and the other reviewers appreciated the quality of writing. Moreover, the AC found that the reviewer lacks expertise in object-centered learning (i.e., he/she has not published any relevant papers to date).

Here is a summary of the reviewers' major concerns and how they are addressed.

- **Lack of novel and essential insights, too obvious conclusions (SC2i, dN2g, neGY):** The reviewers are not satisfied with the response. SC2i and neGY were supportive of this paper but gave borderline accept due to this issue, while it is the main objection of dN2g, the negative reviewer (again, except ixZS).
- **Incremental novelty of the proposed method (SC2i, neGY)**: The reviewers are not satisfied with the response, the AC found it convincing though as mentioned in the meta-review (in short, the proposed method is more like a bag of engineering tricks but the AC believes it is still valuable and should be widely known by researchers in the field.)
- **Missing analysis or solution for the scalability issue of the proposed method (SC2i)**: Well addressed; the focus of this paper is to reveal and report this issue of existing object-centric learning methods, not to resolve it, which is out of the scope of this paper.
- **Small scales of the datasets used in the paper (SC2i)**: The authors admitted, but also argued that the pretraining is not the goal of this paper or what this paper encourages; then this issue may not be a ground for rejection.
- **Unclear motivation and advantage of the learning rate scheduling strategy (SC2i)**: Successfully addressed, according to the reviewer.
- **No design element for zero-shot transfer presented although the benchmark targets zero-shot trasfer (dN2g)**: The rebuttal well addressed this issue by clarifying the goal of zero-shot transfer in thie paper. Also this was not a major objection of the reviewer.
- **Issues on the evaluation protocol (neGY)**: The reviewer mentioned that the use of pretrained DINO features makes unclear if the zero-shot capability comes from object-centric learning strategies or the features, and that the paper needs to investigate more downstream tasks for evaluation. The authors' response well addressed these concerns.

---

### Decision · Program_Chairs · 2025-01-22

Accept (Poster)